

# Phases of cold holographic QCD: Baryons, pions and rho mesons

Nicolas Kovensky[1*], Aaron Poole[2†] and Andreas Schmitt[3‡]

**1** Institut de Physique Théorique, Université Paris Saclay, CEA, CNRS,
Orme des Merisiers, 91191 Gif-sur-Yvette CEDEX, France
**2** Department of Physics and Research Institute of Basic Science,
Kyung Hee University, Seoul 02447, Korea
**3** Mathematical Sciences and STAG Research Centre, University of Southampton,
Southampton SO17 1BJ, United Kingdom

★ nicolas.kovensky@ipht.fr , † apoole@khu.ac.kr , ‡ a.schmitt@soton.ac.uk

## Abstract

We improve the holographic description of isospin-asymmetric baryonic matter within the Witten-Sakai-Sugimoto model by accounting for a realistic pion mass, computing the pion condensate dynamically, and including rho meson condensation by allowing the gauge field in the bulk to be anisotropic. This description takes into account the co-existence of baryonic matter with pion and rho meson condensates. Our main result is the zero-temperature phase diagram in the plane of baryon and isospin chemical potentials. We find that the effective pion mass in the baryonic medium increases with baryon density and that, as a consequence, there is no pion condensation in neutron-star matter. Our improved description also predicts that baryons are disfavored at low baryon chemical potentials even for arbitrarily large isospin chemical potential. Instead, rho meson condensation sets in on top of the pion condensate at an isospin chemical potential of about $9.4\,m_\pi$. We further observe a highly non-monotonic phase boundary regarding the disappearance of pion condensation.

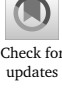

# 1 Introduction and conclusions

## 1.1 Context

Phases of Quantum Chromodynamics (QCD) at non-zero, and not asymptotically large, baryon chemical potential $\mu_B$ are notoriously difficult to understand from first principles. In contrast, at non-zero isospin chemical potential $\mu_I$ lattice QCD can be employed [1–5]. In this paper we connect – within a single model – the mesonic phases at $\mu_I \gg \mu_B$ with nuclear matter at $\mu_B \gg \mu_I$ and discuss the phases in between, where baryons coexist with meson condensates.

To this end, we employ the Witten-Sakai-Sugimoto (WSS) model [6–8], which is a certain realization of the gauge/gravity correspondence [9,10], based on the holographic principle. Holographic models have been employed to understand aspects of QCD matter that are non-perturbative in nature, for instance in the context of heavy-ion collisions [11,12]. More recently, they have also increasingly been employed to understand dense baryonic matter in the context of neutron stars [13–22]. Ideally, these calculations would make use of a string dual of QCD. Such a dual, however, is currently unknown, and thus holographic models can provide results for theories similar, but not identical, to QCD. The WSS model is a top-down approach originating from type-IIA string theory. In a certain, albeit inaccessible, limit it is dual to QCD at a large number of colors $N_c$, and it has proven useful in the discussion of meson, baryon, and glueball spectra and their interactions [23–26] as well as various properties of QCD matter [27,28].

Holographic baryonic matter has been studied in different approximations within the (top-down) WSS model [29–36], besides other holographic (bottom-up) approaches [37,38]. Our main motivation is to improve the recently developed holographic description of isospin-asymmetric baryonic matter [39], which despite its simplicity and shortcomings was shown to yield realistic neutron stars in agreement with astrophysical data [40,41]. Following the basic setup of these studies, we will work in the background geometry corresponding to the con-

fined phase, use the Yang-Mills approximation to the Dirac-Born-Infeld action, consider two flavors, and employ the so-called "homogeneous ansatz", where the gauge fields in the bulk are assumed to only depend on the holographic coordinate [30, 32, 34]. This is in contrast to the single-baryon solution of the model, which is a localized instanton in position space as well as in the holographic direction, and whose direct generalization to a many-baryon system is an alternative – somewhat more difficult – description [31–33, 35].

Our main novelties are as follows. Firstly, we include a non-zero pion mass. Due to the geometry of the WSS model, this is more difficult compared to other holographic setups. We shall therefore follow the effective description brought forward in Refs. [42–44] and evaluated in the context of the QCD phase structure (without isospin chemical potential) in Refs. [36, 45]. This improvement allows us to match our holographic results to chiral perturbation theory, compute the onset of pion condensation in the medium of baryonic matter, and determine the pion condensate dynamically throughout the phase diagram. Secondly, when solving the equations of motion for the bulk gauge fields we use a more general ansatz compared to Ref. [39], where an isotropic approximation was employed for the non-abelian spatial components, and the abelian spatial components were assumed to vanish. The more general approach, in particular the anisotropy in the gauge field configuration, allows for a continuous transition to phases with rho meson condensation, which were previously discussed in the WSS model only in the absence of baryons [46]. Nevertheless, even in our generalized ansatz we employ a simplification by keeping the non-abelian gauge fields diagonal – locking spatial indices with those indicating the orientation in isospin space. We will argue that this is a good approximation for our purposes since our main results are tightly constrained by the regimes where the diagonal ansatz does provide an exact solution to our equations of motion.

We use our improved description to address various open questions and make model predictions for uncharted territory of the QCD phase diagram. In the region of relatively small isospin asymmetry, $\mu_I \ll \mu_B$, we may ask under which conditions a pion condensate coexists with baryonic matter. The possibility of a $p$-wave pion condensate in dense nuclear matter and thus the interior of neutron stars was pointed out long ago [47–51], while an $s$-wave condensate is considered less likely due to the repulsive pion-neutron interaction, although it cannot be ruled out completely [52]. In the opposite limit, where $\mu_I \gg \mu_B$, one may ask whether baryonic states with large isospin number are populated or whether the entire isospin density is generated by mesons. It was conjectured that there is a continuity from pion condensation at low $\mu_I$ to Cooper pairs with the same quantum number as the pions at asymptotically large $\mu_I$ without the appearance of baryons [53]. Recent studies on the lattice have shed some light on the equation of state at non-zero isospin chemical potential [3, 5], with results including a small baryon chemical potential obtained via Taylor expansion [4], however without conclusive evidence about the (non-)appearance of baryons. These results have been used to speculate about the existence of pion stars [54–56]. Finally, we can extend our results to the region with a sizable $\mu_B$ (say comparable to values at the center of neutron stars) and $\mu_I$ larger than found in any known astrophysical environment. Although this region might currently be only of academic interest, it may help inform our understanding of the phases in the regions which *are* accessible experimentally or astrophysically.

## 1.2 Main result

Our main result is the phase diagram in Fig. 1. We will explain the calculation leading to this diagram in the main text and give our conclusions here. The WSS model in the form used here has three parameters, the Kaluza-Klein mass $M_{KK}$, the 't Hooft coupling $\lambda$, and the pion mass $m_\pi$. For Fig. 1 we have fitted them to reproduce the physical pion mass $m_\pi \simeq 140$ MeV, the mass of the rho meson $m_\rho \simeq 776$ MeV, and the pion decay constant $f_\pi \simeq 92.4$ MeV, following the fit already discussed in the original works by Sakai and Sugimoto [7,8], resulting

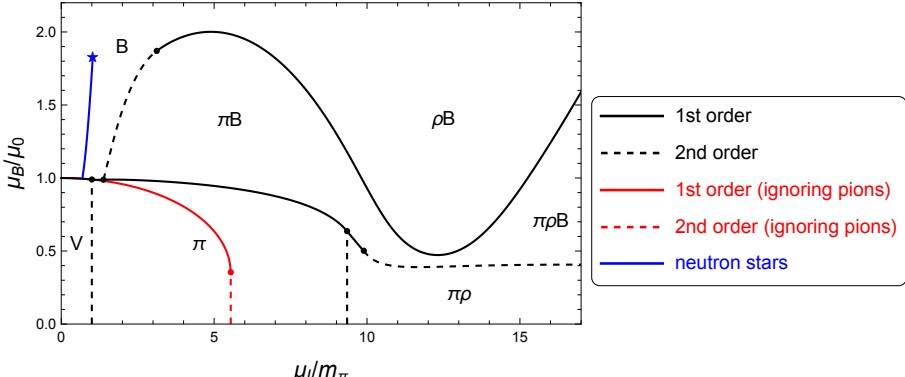

Figure 1: Zero-temperature phase diagram in the plane of baryon and isospin chemical potentials, in units of the onset chemical potential of isospin-symmetric nuclear matter $\mu_0$ and the pion mass in the vacuum $m_\pi$. The various phases are the vacuum (V), the pion-condensed phase ($\pi$), the phase where pion and rho condensates coexist ($\pi\rho$), the purely baryonic phase (B) and the phase where baryonic matter coexists with a pion condensate ($\pi$B). In both baryonic phases a rho meson condensate develops at large $\mu_I$, the resulting $\rho$B and $\pi\rho$B phases are not distinguished by a phase transition in our approximation. The red curves show, for comparison, the phase structure in the absence of pion condensation. They separate the vacuum from the B (and $\rho$B) phase. The blue curve is $\beta$-equilibrated, electrically neutral matter relevant for neutron stars, with the marker at the endpoint indicating the chemical potentials in the center of the maximum mass star.

in $M_{\mathrm{KK}} = 949\,\mathrm{MeV}$, $\lambda = 16.6$. As in most previous applications of the model, this implies an extrapolation from the regime of very large $\lambda$, where the classical gravity approximation in the bulk is valid, down to smaller, but not perturbatively small, coupling strengths.

The main observations are as follows.

- *Rho meson condensation with pions switched off.* The phase transition lines with pion condensation artificially turned off (red) show the usual first-order baryon onset at $\mu_I = 0$ from the vacuum to isospin-symmetric nuclear matter (the black and red lines are on top of each other for the V-B transition). At $\mu_B = 0$, we have a second-order transition at $\mu_I = 776\,\mathrm{MeV} \simeq 5.5\,m_\pi$, which corresponds to the mass of the rho meson. The resulting phase with rho meson condensation is necessarily anisotropic because the rho meson is a vector meson. Within our approximation, this phase is continuously connected to the baryonic phase. The reason is that in our approximation isospin and position space are coupled and thus any non-zero $\mu_I$ creates an anisotropy in our solution. This anisotropy is small at small $\mu_I$ (purely baryonic phase) and becomes maximal – in the absence of pion condensation – only at exactly $\mu_B = 0$ (pure rho meson phase).

- *No baryons at small $\mu_B$ for any $\mu_I$.* At sufficiently small $\mu_B$, the pion-condensed phase ($\pi$) is superseded by a phase where pion and rho condensates coexist. The second-order onset of rho meson condensation in the pionic medium is at $\mu_I \simeq 9.4\,m_\pi$. This is in agreement with Ref. [46]. This reference worked in the chiral limit, but nevertheless observed approximately the same critical chemical potential since the pion mass is negligible in this regime. The second, more fundamental, difference to this reference is that our calculation shows that the $\pi\rho$ phase is indeed preferred over a baryonic phase. *A priori*, it is conceivable that baryons appear for large $\mu_I$ even though $\mu_B$ is small because they may contribute to the isospin density. In fact, this was observed in Ref. [39] within

the same model, where the approximate isotropic solution was extrapolated to large $\mu_I$. Here, allowing for anisotropic solutions, we conclude that our previous approximation underestimated the free energy of the baryonic phase (i.e., made the highly asymmetric baryonic phase more favorable), and that the model actually predicts a finite band at low $\mu_B$ that is purely mesonic. This band only exists if pions are taken into account.

- *Nuclear matter delays pion condensation.* In our holographic baryonic matter, the onset of pion condensation occurs at a *larger* critical chemical potential $\mu_I$ than in the vacuum. This observation is in qualitative agreement with an earlier holographic study using a bottom-up approach [37]. (Due to this shift in the onset, there is a small segment of a direct transition from the $\pi$ to the B phase.) As a consequence, we find that nuclear matter under neutron star conditions – requiring electroweak equilibrium and electric charge neutrality – does not exhibit pion condensation, in accordance with the predicted absence of $s$-wave condensation from more traditional approaches. This provides another improvement on Ref. [39], where due to the absence of a pion mass, pion condensation set in immediately at $\mu_I > 0$. It validates *a posteriori* the assumption used in our construction of neutron stars from holography [40], where pion condensation was omitted. We have used the construction of this reference to compute the chemical potentials in the center of the most massive star, which is indicated in the phase diagram.[1] (This calculation includes the holographic construction of the crust of the star and thus, following Ref. [40], uses the surface tension of nuclear matter as an additional input parameter, which we have set to $\Sigma = 1\,\mathrm{MeV/fm^2}$.) Moreover, for all $\mu_I$ we consider, the pion condensate vanishes for sufficiently large $\mu_B$. This can only be observed in our improved description with anisotropic baryons and a non-zero pion mass.

- *Curious behavior for large $\mu_B$ and $\mu_I$.* The onset of pion condensation in the baryonic medium eventually turns into a first-order transition. At very large $\mu_I$ and $\mu_B$ there is a very pronounced non-monotonicity in this B-$\pi$B transition. We are not aware of any other model calculation to compare this result with and there is no experimental or astrophysical data for this extreme regime. The transition curve starts to become non-monotonic at a value of $\mu_I$ around the rho meson mass and can thus be related to rho meson admixtures in the B and $\pi$B phases, denoted by $\rho$B and $\pi\rho$B in the phase diagram. We will discuss the dependence of the phase structure on the 't Hooft coupling $\lambda$ and see that the non-monotonicity disappears for large values of $\lambda$, where our classical gravity approximation is more reliable. In any case, for extremely large chemical potentials we have to interpret the results with care since our probe brane approximation breaks down and, moreover, in real-world QCD we expect deconfinement and chiral transitions, which in our setup are absent.

- *Parameter dependence.* The parameter fit used here reproduces the basic mesonic properties in the vacuum but is in tension with saturation properties of symmetric nuclear matter. It predicts an onset chemical potential $\mu_0 \simeq 1430\,\mathrm{MeV}$ and a saturation density $n_0 \simeq 0.43\,\mathrm{fm^{-3}}$, compared to the physical values $\mu_0 \simeq 923\,\mathrm{MeV}$ and $n_0 \simeq 0.15\,\mathrm{fm^{-3}}$. We will show that it is possible to choose the parameters to reproduce $\mu_0$, $n_0$, $m_\rho$, and $m_\pi$ to very good accuracy at the cost of making $f_\pi$ smaller than the physical value. This choice requires $\lambda = 7.8$. It does not change any of our main conclusions in the regime $\mu_B \gg \mu_I$. It does, however, change the structure of the phase diagram at $\mu_I \gg \mu_B$. Since in this regime it is more important to work with a correct pion decay constant than with correct properties of symmetric nuclear matter, we have chosen $\lambda = 16.6$ for Fig. 1: it

---

[1]Pion stars (with electric neutrality and electroweak equilibrium) are located on the $\mu_I$ axis with maximal values of $\mu_I$ very close to the pion mass, see Table I of Ref. [55].

is our best prediction for the $\mu_I \gg \mu_B$ regime and, as we will show, it does not distort the conclusions for the $\mu_B \gg \mu_I$ regime, where the phase structure appears to be robust against moderate variations of the parameters.[2]

The discussion about the best "global" fit of the parameters gets more complicated if astrophysical data for neutron stars are taken into account. At the given $M_{\text{KK}}$, which is used to fit the rho meson mass, both values of $\lambda$ violate astrophysical constraints, either the maximal neutron star mass is too small ($\lambda = 16.6$) or the tidal deformability of the 1.4 solar mass star is too large ($\lambda = 7.8$). It is possible to fulfill all constraints from neutron stars even for a physical rho meson mass by choosing an appropriate $\lambda$ in between these values [40,41]. For our present purpose it is sufficient to conclude that there is no pion condensate in neutron stars for values of $\lambda$ that (approximately) reproduce real-world mesonic and/or baryonic properties.

## 1.3  Outlook

Our work is a step towards more realistic holographic baryonic matter, in particular clarifying the connection to meson condensation at non-zero isospin chemical potential. Various improvements and extensions are possible in future work. One may perform an analogous calculation in the deconfined geometry of the model, which would give rise to a non-trivial temperature dependence, and thus our phase diagram can in principle be generalized by adding temperature as a third axis. This might be of relevance for a more detailed comparison with lattice results at $\mu_B = 0$, and in the astrophysical context it would provide an equation of state applicable for conditions in neutron star mergers, where zero temperature is no longer a good approximation. The deconfined geometry also provides a deviation from chiral perturbation theory already in the purely pionic phase [39], which is a further motivation to include a non-zero pion mass in this computationally more difficult setting.

Furthermore, it is known that the symmetry energy of holographic nuclear matter in the present approach is unphysically large due to the classical treatment of the isospin spectrum, which is continuous in the large-$N_c$ limit [39]. Recently, a quantization of the homogeneous ansatz has been proposed [57] and it would be interesting to study the effect of this improvement on our phase diagram. Our approach also ignores the possibility of an anisotropic pion condensate, which would be desirable to include for a comparison with predictions of $p$-wave condensation in neutron star matter. This could be included in the holographic setup by a different choice of boundary conditions, as done in Refs. [58,59] in the presence of a magnetic field. In the neutron star context it would also be very interesting to consider kaon condensation [60,61] instead of or possibly in coexistence with pion condensation. This would require the generalization to three flavors which, as a first step, could be done in the mesonic sector only, before including baryons with strangeness in a second step.

## 1.4  Structure of the paper

Our paper is organized as follows. We present our holographic setup and discuss its relation to chiral perturbation theory in Sec. 2. Important ingredients of the setup compared to earlier work are the chiral rotation discussed in Sec. 2.1 and the mass correction discussed in Sec. 2.2. In Sec. 3 we introduce our holographic baryonic matter and derive the equations of motion, the free energy, and the minimization conditions for the various dynamical parameters. These

---

[2]In v1 on the arXiv, Fig. 1 shows a phase diagram for $\lambda = 7.8$ that is qualitatively identical to the one shown here in all regimes, in contrast to what we just argued. The reason is that in this first version we ignored the abelian spatial components of the gauge fields (we thank Lorenzo Bartolini and Matti Järvinen for drawing our attention to this point). In the present version these components are included, and now choosing $\lambda = 16.6$ for the reasons just explained keeps the main conclusions unaltered compared to v1.

results are used in Sec. 4 to discuss the different solutions and their interpretation as distinct physical phases. Sec. 5 is devoted to the numerical evaluation: We explain the parameter fits in Sec. 5.1, and while the main results have already been discussed in the introduction, we present a more detailed analysis in Secs. 5.2 – 5.5.

## 2 Setup and holographic pion condensation

We start by reviewing pion condensation at non-zero isospin chemical potential within chiral perturbation theory [62,63], and how it is reproduced from the holographic perspective. This will be useful to establish notation, to explain the particular chiral rotation which we shall apply for convenience, and to introduce the mass term, which will be added to our holographic action to effectively account for a non-zero pion mass.

### 2.1 Pion condensation from the chiral Lagrangian

Two-flavor QCD in the massless limit is invariant under the global chiral symmetry group $U(2)_L \times U(2)_R$. In the real world, where quarks are massive, this symmetry is only approximate (and the axial $U(1)_A \subset U(2)_L \times U(2)_R$ is broken due to the chiral anomaly). At low energies, where the perturbative description in terms of the fundamental quark and gluon degrees of freedom becomes inapplicable, chiral symmetry is spontaneously broken down to its diagonal subgroup, $U(2)_L \times U(2)_R \to U(2)_{L+R}$. The low-energy effective theory for the associated pseudo-Goldstone bosons is formulated in terms of the pion and mass matrices $\Sigma, M \in U(2)$, which transform under the chiral group as

$$\Sigma' = g_L \Sigma g_R^\dagger, \qquad M' = g_L M g_R^\dagger, \tag{1}$$

where $(g_L, g_R) \in U(2)_L \times U(2)_R$. To leading order in derivatives and in the pion mass, the chiral Lagrangian reads [64,65]

$$\mathcal{L} = \frac{f_\pi^2}{4} \mathrm{Tr}[D_\mu \Sigma^\dagger D^\mu \Sigma] + \frac{f_\pi^2 B}{2} \mathrm{Tr}[M \Sigma^\dagger + \Sigma M^\dagger], \tag{2}$$

where $f_\pi$ is the pion decay constant. The mass matrix contains up and down quark masses, $M = \mathrm{diag}(m_u, m_d)$, while $B$ is proportional to the chiral condensate and can be expressed in terms of pion mass and quark mass $m_q \simeq m_u \simeq m_d$ via $m_\pi^2 = 2m_q B$. In the absence of electromagnetism, the covariant derivative is

$$D^\mu \Sigma = \partial^\mu \Sigma - i \frac{\mu_I}{2} [\tau_3, \Sigma] \delta_0^\mu. \tag{3}$$

Here and in the following we denote the Pauli matrices by $\tau_a$ ($a = 1, 2, 3$). The isospin chemical potential is normalized such that, as we will see shortly, pion condensation occurs at $\mu_I = m_\pi$. The baryon chemical potential comes together with the unit matrix in flavor space and thus drops out of the covariant derivative, i.e., in this section, where we only discuss the mesonic sector, the results do not depend on $\mu_B$. (In the presence of a magnetic field, the chiral anomaly gives rise to a $\mu_B$ dependence of the chiral Lagrangian [66,67].)

The pion field can be parameterized as

$$\Sigma = \frac{1}{f_\pi} (\sigma \mathbb{1} + i \pi_a \tau_a), \tag{4}$$

where the massive mode $\sigma^2 = f_\pi^2 - \pi_a \pi_a$ is frozen and $\pi_a$ are the three pionic degrees of freedom. Alternatively, we can parameterize

$$\sigma = f_\pi \cos\psi \cos\theta\,, \tag{5a}$$

$$\pi_1 = f_\pi \cos\psi \sin\theta \cos\alpha\,, \tag{5b}$$

$$\pi_2 = f_\pi \cos\psi \sin\theta \sin\alpha\,, \tag{5c}$$

$$\pi_3 = f_\pi \sin\psi\,. \tag{5d}$$

Then, in the static, homogeneous limit and subtracting the vacuum contribution, the free energy density derived from the Lagrangian (2) is

$$\Omega = -\frac{\mu_I^2}{2}(\pi_1^2 + \pi_2^2) - f_\pi m_\pi^2(\sigma - f_\pi) = -\frac{\mu_I^2}{2}f_\pi^2 \sin^2\theta \cos^2\psi - f_\pi^2 m_\pi^2(\cos\psi \cos\theta - 1)\,. \tag{6}$$

Upon minimizing $\Omega$ with respect to $\psi$ and $\theta$, one finds that the ground state for $\mu_I < m_\pi$ is the vacuum $\theta = \psi = 0$, where the chiral field is $\Sigma = \mathbb{1}$ and the free energy density is $\Omega = 0$. On the other hand, when $\mu_I > m_\pi$, charged pion condensation becomes favored, with a vanishing neutral pion condensate $\psi = \pi_3 = 0$, and

$$\cos\theta = \frac{m_\pi^2}{\mu_I^2}\,, \qquad \Omega = -\frac{f_\pi^2 \mu_I^2}{2}\left(1 - \frac{m_\pi^2}{\mu_I^2}\right)^2\,, \qquad n_I = -\frac{\partial\Omega}{\partial\mu_I} = f_\pi^2 \mu_I\left(1 - \frac{m_\pi^4}{\mu_I^4}\right)\,, \tag{7}$$

where $n_I$ is the isospin density. For later use we shall denote the pion matrix with vanishing neutral pion condensate by

$$\Sigma_0 = \cos\theta\, \mathbb{1} + i\sin\theta(\tau_1 \cos\alpha + \tau_2 \sin\alpha)\,. \tag{8}$$

Here, in the purely mesonic scenario, the charged pion condensate is given by Eq. (7), while it will be determined dynamically in our holographic calculation in the presence of baryons. In either case, the system is degenerate with respect to the angle $\alpha$, which corresponds to the $U(1)_{L+R}$ under which the Lagrangian is invariant even in the presence of an isospin chemical potential. We could therefore set $\alpha$ to any convenient value at this point, but we will keep it unspecified for now in order to check explicitly the degeneracy with respect to $\alpha$ within the full holographic calculation.

For reasons explained in Refs. [39, 46, 58] and that will become clear below, in our holographic study it will be useful to apply a chiral rotation (1) such that

$$\Sigma_0' = g_L \Sigma_0 g_R^\dagger = \mathbb{1}\,. \tag{9}$$

We choose

$$g_R = g_L^\dagger \equiv g\,, \tag{10}$$

such that $g^2 = \Sigma_0$, which is satisfied for instance by

$$g = \cos\frac{\theta}{2}\,\mathbb{1} + i\sin\frac{\theta}{2}(\tau_1 \cos\alpha + \tau_2 \sin\alpha)\,. \tag{11}$$

This defines separate left- and right-handed transformations for $U(2)_L$ and $U(2)_R$ matrices. Since in either case the matrices can be written as linear combinations of the Pauli matrices (and the unit matrix), we compute the transformations on $\boldsymbol{\tau} = (\tau_1, \tau_2, \tau_3)$ from (11),

$$\boldsymbol{\tau}'_{R/L} = \begin{pmatrix} \cos\theta \sin^2\alpha + \cos^2\alpha & \sin\alpha \cos\alpha(1 - \cos\theta) & \pm \sin\alpha \sin\theta \\ \sin\alpha \cos\alpha(1 - \cos\theta) & \cos\theta \cos^2\alpha + \sin^2\alpha & \mp \cos\alpha \sin\theta \\ \mp \sin\alpha \sin\theta & \pm \cos\alpha \sin\theta & \cos\theta \end{pmatrix} \boldsymbol{\tau}\,, \tag{12}$$

where $\tau'_{a,R/L} = g_{R/L} \tau_a g^\dagger_{R/L}$. In the absence of pion condensation, $\theta = 0$, both left- and right-handed transformations become the identity.

In our holographic model, the isospin chemical potential will be introduced through boundary conditions for the temporal components of the gauge fields on the left- and right-handed branes. Therefore, if we want to perform the calculation conveniently in the frame where $\Sigma'_0 = \mathbb{1}$, the pion condensate will enter the boundary conditions via the transformation (12).

The choice (10) is clearly not unique. In particular, it differs from the one used in Refs. [39,46,58]. These references all worked in the chiral limit, and the transformation used there was defined by $g_R = \mathbb{1}$ and $g_L = \Sigma^\dagger_0$. In the chiral limit, the pion condensate is maximal, $\cos \theta = 0$, for all $\mu_I$, as can be seen for instance from Eq. (7) (as we shall see later, this is true even in the presence of baryons). In this case, this alternative transformation gives the identity in the right-handed sector and $g_L \tau_3 g^\dagger_L = -\tau_3$ for all $\alpha$ (with a rotation in the $\tau_1$-$\tau_2$ sector that depends on $\alpha$) in the left-handed sector. Hence, this choice is particularly simple. However, generalizing this transformation to the case of a non-zero pion mass leads to asymmetric values in left- and right-handed sectors, rendering the holographic computation more cumbersome. This is avoided by the symmetric choice (10), which will allow us to work with symmetric or antisymmetric boundary conditions for any value of $\theta$.

## 2.2 Holographic setup including effective mass term

The WSS model [6–8] is a top-down string-theoretical construction describing the near-horizon geometry of a non-supersymmetric configuration sourced by $N_c$ D4-branes. The flavor sector is included by adding $N_f$ D8-$\overline{\text{D8}}$-brane pairs which we assume to be maximally separated along a compact circle $X_4 \sim X_4 + 2\pi M^{-1}_{\text{KK}}$. The flavor branes thus follow geodesics, hence their embedding does not have to be determined dynamically. In the so-called confined geometry, the induced metric on the flavor branes takes the form

$$ds^2 = \left(\frac{U}{R}\right)^{3/2} \left(dX_0^2 + d\boldsymbol{X}^2\right) + \left(\frac{R}{U}\right)^{3/2} \left[\frac{dU^2}{f(U)} + U^2 d\Omega_4^2\right], \qquad f(U) = 1 - \frac{U^3_{\text{KK}}}{U^3}. \tag{13}$$

Here, $U$ is the coordinate for the radial holographic direction, $d\Omega_4^2$ describes the unit 4-sphere, $R$ is the background curvature radius, which is related to the string length $\ell_s$, the 't Hooft coupling and the Kaluza-Klein mass via $R^3 = \lambda \ell_s^2/(2M_{\text{KK}})$, and $U_{\text{KK}} = 2\lambda M_{\text{KK}} \ell_s^2/9$ is the location where the $X_4$ direction caps off. Four-dimensional Euclidean space-time is given by $(X_0, \boldsymbol{X})$, where the temporal component $X_0 \sim X_0 + T^{-1}$ is compact in the presence of a non-zero temperature $T$. In this version of the model, large $N_c$ effects render the flavor physics temperature independent, so that $T$ will appear only as an overall factor in the free energy.

The action on the flavor branes is composed of the Dirac-Born-Infeld (DBI) and Chern-Simons (CS) contributions, together with our effective mass term,

$$S = S_{\text{DBI}} + S_{\text{CS}} + S_{\text{m}}. \tag{14}$$

We now discuss each of these terms separately.

### 2.2.1 Yang-Mills and Chern-Simons contributions

The DBI action is

$$S_{\text{DBI}} = 2T_8 V_4 \int d^4X \int_{U_{\text{KK}}}^\infty dU e^{-\phi} \text{STr} \sqrt{\det(g + 2\pi\alpha'\mathcal{F})}, \tag{15}$$

where $T_8 = 1/\left[(2\pi)^8 \ell_s^9\right]$ is the D8-brane tension, $V_4 = 8\pi^2/3$ is the volume of the 4-sphere, $e^\phi = g_s (U/R)^{3/4}$ is the dilaton with the string coupling $g_s = \lambda/(2\pi N_c M_{\text{KK}} \ell_s)$, and $\alpha' = \ell_s^2$.

The prefactor 2 accounts for the two halves of the connected flavor branes. The metric $g$ is given by Eq. (13), and the field strength $\mathcal{F}$ can be expressed in terms of the world-volume gauge field $\mathcal{A}$ as

$$\mathcal{F}_{\mu\nu} = \partial_\mu \mathcal{A}_\nu - \partial_\nu \mathcal{A}_\mu + i[\mathcal{A}_\mu, \mathcal{A}_\nu], \tag{16}$$

with $\mu, \nu \in \{0, 1, 2, 3, U\}$. We work with $N_f = 2$, and, following the convention of Ref. [39], introduce the dimensionless coordinates

$$u = \frac{U}{R(M_{\text{KK}}R)^2}, \qquad x_0 = \lambda_0 M_{\text{KK}} X_0, \qquad x_i = M_{\text{KK}} X_i, \tag{17}$$

where $i = 1, 2, 3$, and where we have abbreviated

$$\lambda_0 \equiv \frac{\lambda}{4\pi}. \tag{18}$$

In these dimensionless units,

$$u_{\text{KK}} = \frac{4}{9}. \tag{19}$$

The corresponding dimensionless gauge fields, decomposed into $U(1)$ and $SU(2)$ parts, are introduced via

$$\mathcal{A}_U = \frac{\hat{A}_u + A_u^a \tau_a}{R(M_{\text{KK}}R)^2}, \qquad \frac{\mathcal{A}_0}{\lambda_0 M_{\text{KK}}} = \hat{A}_0 + A_0^a \tau_a, \qquad \frac{\mathcal{A}_i}{M_{\text{KK}}} = \hat{A}_i + A_i^a \tau_a. \tag{20}$$

Accordingly, we introduce abelian and non-abelian components of the dimensionless field strengths

$$F_{\mu\nu} = \hat{F}_{\mu\nu} + F_{\mu\nu}^a \tau_a, \tag{21}$$

with

$$\hat{F}_{\mu\nu} = \partial_\mu \hat{A}_\nu - \partial_\nu \hat{A}_\mu, \qquad F_{\mu\nu}^a = \partial_\mu A_\nu^a - \partial_\nu A_\mu^a - 2\epsilon_{abc} A_\mu^b A_\nu^c. \tag{22}$$

Moreover, we shall from now on assume all fields to be independent of Euclidean space-time, i.e., they only depend on the holographic coordinate $u$. Then, the space-time integration becomes trivial and simply yields a prefactor $V/T$, where $V$ is the 3-volume of our system. We work with the Yang-Mills (YM) approximation to the DBI action, keeping only terms of second order in the field strength. In this approximation, there is no ambiguity in calculating the action, the symmetrized trace in Eq. (15) is identical to an ordinary trace. The result is

$$S_{\text{DBI}} \simeq S_{\text{YM}} = \frac{\mathcal{N}}{2} \frac{V}{T} \int_{u_{\text{KK}}}^\infty du \left[ u^{5/2} \sqrt{f} \left( \text{Tr}[F_{0u}^2] + \frac{\text{Tr}[F_{iu}^2]}{\lambda_0^2} \right) + \frac{1}{u^{1/2}\sqrt{f}} \left( \text{Tr}[F_{0i}^2] + \frac{\text{Tr}[F_{ij}^2]}{2\lambda_0^2} \right) \right], \tag{23}$$

where we have abbreviated

$$\mathcal{N} = \frac{N_c M_{\text{KK}}^4 \lambda_0^3}{6\pi^2}. \tag{24}$$

The CS action can be written in terms of abelian and non-abelian components as [23, 58]

$$S_{\text{CS}} = -i\frac{\mathcal{N}}{2\lambda_0^2} \frac{V}{T} \int_{u_{\text{KK}}}^\infty du \left\{ \frac{3}{2} \hat{A}_\mu \left( F_{\nu\rho}^a F_{\sigma\lambda}^a + \frac{1}{3} \hat{F}_{\nu\rho} \hat{F}_{\sigma\lambda} \right) \right.$$

$$\left. + 2\partial_\mu \left[ \hat{A}_\nu \left( F_{\rho\sigma}^a A_\lambda^a + \frac{1}{4} \epsilon_{abc} A_\rho^a A_\sigma^b A_\lambda^c \right) \right] \right\} \epsilon^{\mu\nu\rho\sigma\lambda}. \tag{25}$$

We will always work in a gauge where $\hat{A}_u = A_u^a = 0$, in which case the pion field is encoded in the boundary conditions for $A_0^a(u)$ [7]. We denote

$$\hat{A}_0(u) \equiv iA(u), \qquad A_0^a(u) \equiv iK_a(u), \tag{26}$$

where the factor $i$ is due to the Euclidean signature of space-time, and where the simplified notation of the temporal gauge fields is introduced to avoid cluttering of indices, in particular in Sec. 3. Then, the YM and CS contributions to the action become

$$S_{\text{YM}} + S_{\text{CS}} = N_f \mathcal{N} \frac{V}{T} \int_{u_{\text{KK}}}^{\infty} du \, \mathcal{L}, \tag{27}$$

with the dimensionless Lagrangian

$$
\begin{aligned}
\mathcal{L} = {}& \frac{u^{5/2}\sqrt{f}}{2}\left(-A'^2 - K_a' K_a' + \frac{\hat{A}_i'\hat{A}_i' + A_i^{a\prime}A_i^{a\prime}}{\lambda_0^2}\right) \\
& + \frac{2}{u^{1/2}\sqrt{f}}\left[K_a A_i^b(K_b A_i^a - K_a A_i^b) + \frac{A_i^a A_j^b(A_i^a A_j^b - A_j^a A_i^b)}{2\lambda_0^2}\right] \\
& + \frac{1}{\lambda_0^2}\epsilon_{ijk}\epsilon_{abc}\left[A(A_i^a A_j^b A_k^c)' - 3\hat{A}_i(K_a A_j^b A_k^c)'\right],
\end{aligned}
\tag{28}
$$

where the last term comes from the CS contribution (only the first term in Eq. (25) contributes), and where prime denotes derivative with respect to $u$. The resulting equations of motion for the temporal components are

$$\left(u^{5/2}\sqrt{f}A'\right)' = -\frac{1}{\lambda_0^2}\epsilon_{ijk}\epsilon_{abc}(A_i^a A_j^b A_k^c)', \tag{29a}$$

$$\left(u^{5/2}\sqrt{f}K_a'\right)' = -\frac{4}{u^{1/2}\sqrt{f}}A_i^b(K_b A_i^a - K_a A_i^b) - \frac{3}{\lambda_0^2}\epsilon_{ijk}\epsilon_{abc}\hat{A}_i' A_j^b A_k^c, \tag{29b}$$

while for the spatial components we have

$$\left(u^{5/2}\sqrt{f}\hat{A}_i'\right)' = -3\epsilon_{ijk}\epsilon_{abc}(A_j^a A_k^b K_c)', \tag{30a}$$

$$
\begin{aligned}
\left(u^{5/2}\sqrt{f}A_i^{a\prime}\right)' = {}& \frac{4\lambda_0^2}{u^{1/2}\sqrt{f}}\left[-K_b(K_b A_i^a - K_a A_i^b) + \frac{A_j^b(A_i^a A_j^b - A_j^a A_i^b)}{\lambda_0^2}\right] \\
& - 3\epsilon_{ijk}\epsilon_{abc}A' A_j^b A_k^c + 6\epsilon_{ijk}\epsilon_{abc}\hat{A}_j' K_b A_k^c.
\end{aligned}
\tag{30b}
$$

### 2.2.2 Mass correction

Finally, we need to specify the effective mass term in the action $S_{\text{m}}$. The WSS model differs from other holographic constructions in that the inclusion of the pion mass cannot be described directly in terms of a separation of the flavor branes from the color branes in a transverse direction. Here we follow the approach of Refs. [42–44], see also Refs. [36, 45, 68–70], where the pion mass is included in an effective way by considering an open Wilson line stretched between the D8- and $\overline{\text{D8}}$-branes. Its expectation value is given by the corresponding worldsheet

action, $\langle\mathcal{O}\rangle = ce^{-S_{\text{WS}}}$ with a constant $c$ and $S_{\text{WS}} = S_{\text{NG}} + S_{\partial}$, and one identifies the (medium dependent) chiral condensate with $\langle\bar{q}q\rangle = -ce^{-S_{\text{NG}}}$. The Nambu-Goto action takes the form

$$S_{\text{NG}} = 2\lambda_0 \int_{u_{\text{KK}}}^{\infty} du\, x_4(u),\tag{31}$$

with $x_4 = M_{\text{KK}}X_4$, in analogy to the spatial coordinates $x_i$ (17). Since we work with maximally separated flavor branes, the embedding function is constant, $x_4 = \pi/2$, and thus $S_{\text{NG}}$ merely contains a constant (infinite) vacuum contribution. Subtracting this vacuum contribution, the exponential containing the Nambu-Goto action is 1. This is in contrast to the case of non-antipodal separation of the flavor branes, where the embedding is medium-dependent and the Nambu-Goto factor gives a non-trivial contribution to the equations of motion [36, 45]. The boundary term $S_{\partial}$ is given by

$$e^{-S_{\partial}} = \exp\left(i\int_{-\infty}^{\infty} dZ\, \mathcal{A}_Z\right) \equiv \Sigma.\tag{32}$$

Here we have introduced the new radial coordinate via

$$u^3 = u_{\text{KK}}^3 + u_{\text{KK}}z^2,\tag{33}$$

and the dimensionful version $Z$ is obtained from $z$ in the same way as $U$ is obtained from $u$ (17). The coordinate $z \in [-\infty, \infty]$ runs from the ultraviolet boundary of the D8-branes to that of the $\overline{\text{D8}}$-branes, with $z = 0$ being at the tip of the connected branes at $u = u_{\text{KK}}$. We will often switch between the coordinates $u$ and $z$ according to which one is more convenient for a given calculation or argument, and when we integrate over $u \in [u_{\text{KK}}, \infty]$ we assume that we are on the $z > 0$ half of the connected branes. The holonomy (32) can be identified with the chiral field $\Sigma$ introduced in the previous subsection in the context of chiral perturbation theory [7]. Therefore, by choosing $c = f_\pi^2 B$, the contribution to the action from the open Wilson line can be matched to the lowest-order mass term of chiral perturbation theory, such that

$$S_{\text{m}} = -\frac{V}{T}\frac{f_\pi^2 B}{2}\text{Tr}[M\Sigma^\dagger + M^\dagger\Sigma],\tag{34}$$

where we have already assumed our system to be homogeneous. With the chiral field $\Sigma = \Sigma_0$ (8) and using the result for the pion decay constant in terms of the parameters of the WSS model [7]

$$f_\pi^2 = \frac{2N_c M_{\text{KK}}^2 \lambda_0}{27\pi^3},\tag{35}$$

the contribution to the action (34) reduces to

$$S_{\text{m}} = -\frac{V}{T}\mathcal{N}N_f\frac{2\bar{m}_\pi^2 \cos\theta}{9\pi},\tag{36}$$

where we have introduced the dimensionless pion mass

$$\bar{m}_\pi \equiv \frac{m_\pi}{\lambda_0 M_{\text{KK}}}.\tag{37}$$

Since the trace is invariant under chiral transformations, the result (36) is independent of whether the rotation (12) is performed on $\Sigma_0$ and $M$ or not. Therefore, the result is also valid for $\mathcal{A}_Z = 0$, for which the chiral field (32) is trivial, but the condensate sits in the rotated matrix $M'$. Within our setup the effective mass term does not contain any gauge field and thus the equations of motion (29) and (30) remain unaltered. However, $S_{\text{m}}$ is not a mere constant

since it contains the charged pion condensate $\theta$ with respect to which we need to minimize the free energy and which enters the gauge fields through the boundary conditions, as we will discuss below. We also note that the identification with the chiral field (32) only works if there is no additional contribution to the holonomy from baryons. We shall comment on this possibility in more detail below Eq. (52).

We can now put together all pieces of our action, Eqs. (27) and (36), to obtain

$$S = \mathcal{N} N_f \frac{V}{T} \left[ \int_{u_{\text{KK}}}^{\infty} du\, \mathcal{L} - \frac{2\bar{m}_\pi^2}{9\pi}(\cos\theta - 1) \right], \tag{38}$$

with the Lagrangian (28). In the vacuum, all gauge fields vanish and thus YM and CS contributions are zero, while the mass term (36) yields a vacuum contribution for $\theta = 0$, which we have subtracted in Eq. (38) to normalize the vacuum pressure to zero.

The grand-canonical potential (= free energy density) is obtained from the on-shell action,

$$\Omega = \frac{T}{V} S \Big|_{\text{on-shell}}, \tag{39}$$

and we shall later work with the dimensionless version

$$\bar{\Omega} = \frac{\Omega}{\mathcal{N} N_f} = \int_{u_{\text{KK}}}^{\infty} du\, \mathcal{L} - \frac{2\bar{m}_\pi^2}{9\pi}(\cos\theta - 1). \tag{40}$$

## 2.3 Reproducing chiral perturbation theory in the WSS model

The action set up in the previous subsection will be used for our main results, where we account for meson condensation and baryonic matter. First, in this subsection, we explain how the results of lowest-order chiral perturbation theory from Sec. 2.1 are reproduced. This will be useful to understand the more complicated calculation in Sec. 3.

In the purely mesonic case it is consistent to set all gauge fields to zero except for the non-abelian temporal components $K_a$. The CS term does not contribute in this scenario, and the Lagrangian (28) reduces to

$$\mathcal{L} = -\frac{u^{5/2}\sqrt{f}}{2} K_a' K_a'. \tag{41}$$

The only non-trivial equation of motion (29b) is

$$\partial_u(u^{5/2}\sqrt{f} K_a') = 0. \tag{42}$$

In terms of the coordinate $z$ of Eq. (33) this yields the solutions

$$K_a(z) = C_a + D_a \arctan\frac{z}{u_{\text{KK}}}, \tag{43}$$

with integration constants $C_a$, $D_a$. According to the usual dictionary of the gauge/gravity correspondence, the boundary value of the temporal component of the gauge field corresponds to the chemical potential of the field theory. The isospin chemical potential corresponds, in the unrotated basis, to the boundary value of the third component $K_3$. More precisely, we work with a vector isospin chemical potential, whose value is the same in the left- and right-handed sectors such that[3] $K_3(z = \pm\infty) = \bar{\mu}_I/2$. Here and in the following we work with dimensionless

---

[3]This convention for $\mu_I$ matches the one of Sec. 2.1, where the onset of pion condensation is at $\mu_I = m_\pi$, but it differs from our previous holographic study [39] by a factor 2.

chemical potentials according to the dimensionless gauge fields of Eq. (20), i.e., we define[4]

$$\bar{\mu}_B = \frac{\mu_B}{\lambda_0 N_c M_{\text{KK}}}, \qquad \bar{\mu}_I = \frac{\mu_I}{\lambda_0 M_{\text{KK}}}. \tag{44}$$

Since we have made the gauge choice $\mathcal{A}_Z = 0$, our chiral field (32) is trivial and it appears we cannot describe pion condensation. This problem can be circumvented by applying the chiral rotation (9), by which we effectively move the pion condensate from the chiral field into the boundary conditions for $K_a$ [39, 46, 58]. In the massless case, the rotation applied in Refs. [39, 46, 58] simply flips the sign of the boundary condition for $K_3$ on the left-handed boundary, and one can easily solve the system with $K_1(z) = K_2(z) = 0$. Since we work with a non-zero pion mass, it is more convenient to apply the rotation (12), as already explained below that equation. This yields the boundary conditions

$$K_1(z \to \pm\infty) = \mp\frac{\bar{\mu}_I}{2} \sin\theta \sin\alpha, \tag{45a}$$

$$K_2(z \to \pm\infty) = \pm\frac{\bar{\mu}_I}{2} \sin\theta \cos\alpha, \tag{45b}$$

$$K_3(z \to \pm\infty) = \frac{\bar{\mu}_I}{2} \cos\theta. \tag{45c}$$

We see that in general all three gauge field components become nonzero. With these boundary conditions the solutions (43) become

$$K_1(z) = -\frac{\bar{\mu}_I}{\pi} \sin\theta \sin\alpha \arctan\frac{z}{u_{\text{KK}}}, \tag{46a}$$

$$K_2(z) = \frac{\bar{\mu}_I}{\pi} \sin\theta \cos\alpha \arctan\frac{z}{u_{\text{KK}}}, \tag{46b}$$

$$K_3(z) = \frac{\bar{\mu}_I}{2} \cos\theta. \tag{46c}$$

Inserting these solutions into the free energy (39), one finds that the free energy density is identical to the one from chiral perturbation theory, Eq. (6) with $\psi = 0$. The dimensionless free energy (40) becomes

$$\bar{\Omega} = -\frac{3 u_{\text{KK}}^{3/2} \bar{\mu}_I^2 \sin^2\theta}{8\pi} - \frac{2\bar{m}_\pi^2}{9\pi}(\cos\theta - 1). \tag{47}$$

After minimizing with respect to $\theta$, the equivalent of Eq. (7) for pion condensate, free energy density, and isospin density in terms of the dimensionless quantities of the holographic calculation is

$$\cos\theta = \frac{\bar{m}_\pi^2}{\bar{\mu}_I^2}, \qquad \bar{\Omega} = -\frac{\bar{\mu}_I^2}{9\pi}\left(1 - \frac{\bar{m}_\pi^2}{\bar{\mu}_I^2}\right)^2, \qquad \bar{n}_I = -\frac{\partial\bar{\Omega}}{\partial\bar{\mu}_I} = \frac{2\bar{\mu}_I}{9\pi}\left(1 - \frac{\bar{m}_\pi^4}{\bar{\mu}_I^4}\right), \tag{48}$$

where we have used Eq. (19). The dimensionless isospin density $\bar{n}_I$ – and, for completeness and later use, baryon density $\bar{n}_B$ – are related to their dimensionful counterparts $n_I$, $n_B$ by

$$n_{B,I} = \frac{N_f \lambda_0^2 M_{\text{KK}}^3}{6\pi^2} \bar{n}_{B,I}. \tag{49}$$

---

[4]The factor $N_c$ in the baryon chemical potential is included because the boundary value of the gauge field corresponds to the *quark* chemical potential, whose dimensionless version, following the notation of Ref. [40], we denote by $\bar{\mu}_B$. This ensures that $\mu_B$ as used in all our physical results is the actual *baryon* chemical potential, which differs by a factor $N_c$ from the quark chemical potential.

# 3 Adding baryons

In this section we introduce baryonic degrees of freedom and explain the setup used for our main results. Baryons in the WSS model are understood as instantonic configurations of the worldvolume gauge fields, such that baryon and instanton numbers are identified. Near the tip of the flavor branes at $z = 0$ a (static) single baryon is well approximated by the classical Belavin-Polyakov-Schwartz-Tyupkin (BPST) configuration [71], where spatial directions are locked to the internal $SU(2)$ directions, $A_i \propto \tau_i$. One can then quantize the collective co-ordinates [72] in order to identify baryonic states with different spin, isospin, and excitation numbers. This procedure was used to compute static properties of the holographic baryonic states [23].

For many-baryon systems the instantonic picture becomes complicated and it is useful to resort to a simpler approximation. Here we follow Refs. [30, 32] and consider a spatially homogeneous distribution of baryonic matter. This can be thought of as highly overlapping instantons and thus we expect this approximation to be accurate at sufficiently large baryon densities. Although it captures part of the relevant physics, the classical treatment of isospin-asymmetric matter we shall employ here induces some unrealistic (large-$N_c$) artifacts. Most notably, it leads to a symmetry energy much larger than in the real world [39]. It was recently proposed to re-introduce the collective coordinate quantization at the level of the homogeneous ansatz to (partially) remedy this deficiency [57]. Here we do not attempt to combine this quantization with our improved ansatz but emphasize that this is a promising idea for future studies.

## 3.1 Improved Ansatz for isospin-asymmetric baryonic matter

Our ansatz for the spatial components of the non-abelian gauge fields is (no summation over $i$)

$$A_i^a(u) = -\frac{\lambda_0}{2} h_i(u) \delta_i^a, \tag{50}$$

where the functions $h_i(u)$ have to be determined dynamically.

This ansatz deserves a few comments. We first observe that, in general, it is *not* consistent to omit all off-diagonal gauge fields $a \neq i$. This can be seen from Eq. (30b): if, for $a \neq i$, off-diagonal gauge fields are set to zero on the right-hand side of this equation, its second and its last term remain non-zero and are of the forms $K_a K_i h_i$ (no summation over $i$) and $\hat{A}'_a K_i h_j$ ($j$ distinct from $a, i$), where $\hat{A}'_a$ can only be consistently set to zero if $K_a h_i h_j$ is zero, on account of Eq. (30a). Therefore, these terms generate a non-zero off-diagonal component $A_i^a$ on the left-hand side (and, in turn, more non-zero terms on the right-hand side are generated). Obviously, if all spatial gauge field components vanish, including the functions $h_i$, there is no inconsistency. This is what we did in the previous section.

There are two more situations in which the off-diagonal components can be switched off consistently, while keeping the diagonal components non-vanishing.

(i) *Isotropic limit:* for $\mu_I = 0$ all functions $K_i(u)$ vanish and a consistent solution exists for $h_1 = h_2 = h_3$. This approximation was extrapolated to non-zero $\mu_I$ (where it is no longer a solution to the equations of motion) in Ref. [39].

(ii) *chiral limit:* in the limit of vanishing pion mass the pion condensate is always maximal, $\sin \theta = 1$, and thus the global rotation employed in Refs. [39, 46, 58] only affects the $\tau_3$ component and the isospin chemical potential does not generate non-zero boundary conditions for $K_1$ and $K_2$. Therefore, these two components can be set to zero such that $K_a K_i h_i = \hat{A}'_a K_i h_j = 0$ even in the presence of non-zero functions $h_i(u)$.

Here, we are interested in the entire $\mu_B$-$\mu_I$ phase diagram, including a physical pion mass and anisotropic solutions. Therefore, strictly speaking, we need to account for off-diagonal gauge components as well, which leads to a very complicated system of equations. It seems we are in trouble. Our solution to this problem is to work with the ansatz (50) as an approximation. The idea is that this ansatz captures the two limits ($i$) and ($ii$) correctly in the regimes where they are valid, while it provides an interpolation for the region in between. We will make this more explicit in Sec. 5.2 with the help of the numerical solutions, showing that the in-between region is in fact very rigidly constrained by the two limits. The benefit is that we can work with a more manageable system of equations.

With the ansatz (50), the Lagrangian (28) can be written as

$$\mathcal{L} = \frac{u^{5/2}}{2\sqrt{f}} \left( g_1 - f A'^2 - f K'_a K'_a + \frac{f}{\lambda_0^2} \hat{A}'_i \hat{A}'_i + g_2 - g_3 \right) - \frac{3\lambda_0}{4} A (h_1 h_2 h_3)'$$

$$- \frac{3}{2} \left[ \hat{A}_1 (h_2 h_3 K_1)' + \hat{A}_2 (h_1 h_3 K_2)' + \hat{A}_3 (h_1 h_2 K_3)' \right], \tag{51}$$

where, generalizing the notation of Ref. [39] to three different functions $h_i(u)$, we have abbreviated

$$g_1 \equiv \frac{f}{4} (h_1'^2 + h_2'^2 + h_3'^2), \tag{52a}$$

$$g_2 \equiv \frac{\lambda_0^2}{4u^3} (h_1^2 h_2^2 + h_1^2 h_3^2 + h_2^2 h_3^2), \tag{52b}$$

$$g_3 \equiv \frac{\lambda_0^2}{u^3} \left[ K_1^2 (h_2^2 + h_3^2) + K_2^2 (h_1^2 + h_3^2) + K_3^2 (h_1^2 + h_2^2) \right]. \tag{52c}$$

The action is still given by Eq. (38) with no further contributions from baryons. In particular, the effective mass term is the same as in the purely mesonic case. The reason is that in our homogeneous ansatz we have $\mathcal{A}_Z = 0$, and thus there is no baryonic contribution to the boundary term (32). This is different in the instantonic picture. In this case, even for a single instanton, the boundary term gives a contribution, which is a correction to the baryon mass from non-zero bare quark masses [69]. Generalizing this calculation to a non-interacting instanton gas would give a correction to our action proportional to $m_\pi n_B$. However, our homogeneous ansatz does not know about the mass of single baryons and thus this contribution is absent. This is also plausible from a physical point of view since we expect this mass correction to be small for large chemical potentials (the $m_\pi n_B$ contribution to the free energy being small compared to terms of order $\mu_B n_B$), and this regime is exactly the one where our homogeneous ansatz is valid.

For a given configuration, the baryon number density is computed from the topological instanton number,

$$n_B = \frac{\lambda_0^3 M_{\text{KK}}^3}{8\pi^2} \int_{-\infty}^{\infty} dz \, \partial_z (h_1 h_2 h_3). \tag{53}$$

Since the functions $h_i$ vanish at the ultraviolet boundaries, $n_B$ is thus only non-zero if the product $h_1 h_2 h_3$ is discontinuous somewhere in the bulk. We assume the discontinuity to sit at $z = 0$ and denote the values of the functions $h_i(z)$ at this point by

$$h_{ic}^{\pm} \equiv h_i(z \to 0^{\pm}). \tag{54}$$

It is conceivable that an instanton system splits into two or more layers in the $z$ direction as the density is increased [31, 33, 34, 36, 73], which would correspond to our discontinuity in

$h_1 h_2 h_3$ moving up in $z$ with its location determined dynamically (and possibly more than one discontinuity appearing). For simplicity we will ignore this possibility in the following.

Our choice for the rotation (10) allows us to work only with functions which are either symmetric or antisymmetric under $z \to -z$. We take the functions $h_i$ to be discontinuous at $z = 0$, and hence write

$$h_{ic}^- = -h_{ic}^+ . \tag{55}$$

With the definition of the dimensionless baryon density (49), we thus obtain from Eq. (53)

$$\bar{n}_B = -\frac{3\lambda_0}{4} h_{1c} h_{2c} h_{3c} , \tag{56}$$

where we have set $h_{ic} \equiv h_{ic}^+$ for notational convenience. The functions $h_i$ are the only ones for which we allow a discontinuity; it is consistent to keep all other functions $A, K_a, \hat{A}_i$ continuous.

As for a single instanton, we require the spatial components of the gauge field to vanish in the ultraviolet,

$$h_i(z \to \pm\infty) = 0 . \tag{57}$$

The ultraviolet boundary condition of the temporal component of the abelian gauge field is given by the baryon chemical potential,

$$A(z \to \pm\infty) = \bar{\mu}_B , \tag{58}$$

while for the non-abelian temporal components $K_a(z)$ we require Eq. (45). The ultraviolet boundary conditions for the abelian spatial components $\hat{A}_i$ are

$$\hat{A}_{1,2}(z \to \pm\infty) = 0 , \qquad \hat{A}_3(z \to \pm\infty) = \pm q_3 , \tag{59}$$

where $q_3$ is the source for a baryon current in the 3-direction. In our approach a nonzero $q_3$ is required in general to suppress the baryon current, which otherwise would be generated due to the connection between isospin and position space directions. In the 1- and 2-directions we may set the sources to zero without generating a current, as expected.

Due to the symmetry of our system we do not expect baryonic matter to be asymmetric in the $\tau_{1,2}$ directions. This corresponds to invariance of our system with respect to rotations by the angle $\alpha$ in Eqs. (45) also in the presence of baryons. Hence, we set

$$h_1(u) = h_2(u) \equiv h(u) . \tag{60}$$

(And, consequently, we denote $h_c \equiv h_{1c} = h_{2c}$.) We find that within our ansatz $\alpha$ does in general induce a difference in the equations of motion for $h_1$ and $h_2$. To respect the symmetry we thus set the constraint $h_1 = h_2$ in the action *before* deriving the equations of motion. This makes a difference in the equation of motion for $h$ and we will show below that in this approach the system *is* invariant under rotations by $\alpha$.

## 3.2 Equations of motion, free energy and minimization conditions

Within this ansatz, the equation of motion for the temporal abelian field (29a) reduces to

$$(u^{5/2}\sqrt{f}A')' = \frac{3\lambda_0}{4}(h^2 h_3)' , \tag{61}$$

which is easily integrated to give

$$A'(u) = \frac{\bar{n}_B Q(u)}{u^{5/2}\sqrt{f(u)}} , \qquad Q(u) \equiv 1 - \frac{h^2(u)h_3(u)}{h_c^2 h_{3c}} . \tag{62}$$

Analogously, Eq. (30a) gives for the spatial abelian components

$$(u^{5/2}\sqrt{f}\hat{A}_1')' = -\frac{3\lambda_0^2}{2}(hh_3K_1)',\qquad(63a)$$

$$(u^{5/2}\sqrt{f}\hat{A}_2')' = -\frac{3\lambda_0^2}{2}(hh_3K_2)',\qquad(63b)$$

$$(u^{5/2}\sqrt{f}\hat{A}_3')' = -\frac{3\lambda_0^2}{2}(h^2K_3)',\qquad(63c)$$

which also can be integrated,

$$\hat{A}_1' = -\frac{3\lambda_0^2 hh_3K_1}{2u^{5/2}\sqrt{f}},\qquad \hat{A}_2' = -\frac{3\lambda_0^2 hh_3K_2}{2u^{5/2}\sqrt{f}},\qquad \hat{A}_3' = -\frac{3\lambda_0^2 h^2K_3}{2u^{5/2}\sqrt{f}}.\qquad(64)$$

In each of the three cases we have set the integration constant to zero. This ensures that $u^{5/2}\sqrt{f}\hat{A}_i'$ vanishes asymptotically in the ultraviolet and thus there are no spatial baryon currents in the boundary theory.

For the non-abelian gauge fields we get

$$(u^{5/2}\sqrt{f}K_1')' = \frac{\lambda_0^2 K_1(h^2+h_3^2)}{u^{1/2}\sqrt{f}} + \frac{9\lambda_0^2 K_1 h^2 h_3^2}{4u^{5/2}\sqrt{f}},\qquad(65a)$$

$$(u^{5/2}\sqrt{f}K_2')' = \frac{\lambda_0^2 K_2(h^2+h_3^2)}{u^{1/2}\sqrt{f}} + \frac{9\lambda_0^2 K_2 h^2 h_3^2}{4u^{5/2}\sqrt{f}},\qquad(65b)$$

$$(u^{5/2}\sqrt{f}K_3')' = \frac{2\lambda_0^2 K_3 h^2}{u^{1/2}\sqrt{f}} + \frac{9\lambda_0^2 K_3 h^4}{4u^{5/2}\sqrt{f}},\qquad(65c)$$

and

$$(u^{5/2}\sqrt{f}h')' = \frac{\lambda_0^2 h}{u^{1/2}\sqrt{f}}\Bigg\{ h^2 + h_3^2 - 2K_1^2 - 2K_2^2 - 4K_3^2$$

$$- \frac{9}{2u^2}\bigg[ h_3^2(K_1^2+K_2^2) + 2h^2 K_3^2 - \frac{2h_3\bar{n}_B Q}{3\lambda_0} \bigg] \Bigg\},\qquad(66a)$$

$$(u^{5/2}\sqrt{f}h_3')' = \frac{\lambda_0^2}{u^{1/2}\sqrt{f}}\Bigg\{ h_3(2h^2 - 4K_1^2 - 4K_2^2) - \frac{9h^2}{u^2}\bigg[ h_3(K_1^2+K_2^2) - \frac{\bar{n}_B Q}{3\lambda_0} \bigg] \Bigg\},\qquad(66b)$$

where we have already inserted the solutions for $A$ and $\hat{A}_i$ [and, for Eq. (66a), set $h_1 = h_2$ before varying the action, as explained below Eq. (60)]. In particular, we see that taking into account the components $\hat{A}_i$ – which were ignored in Ref. [39] – does not render the calculation much more difficult: since their equations of motion can be integrated analytically, they only generate additional terms in the other equations, not additional non-trivial equations.

Equations (65) and (66) can obviously not be integrated as easily as Eq. (61) and will have to be solved numerically. Simultaneously, we will need to determine the charged pion condensate $\theta$ and the boundary values $h_c, h_{3c}$ by minimizing the free energy. To this end, we take the derivative of the free energy with respect to a generic variable $x \in \{\theta, h_c, h_{3c}, \bar{\mu}_B, \bar{\mu}_I, q_3, \alpha\}$, with all other variables in this set kept fixed. Here we have included the chemical potentials

in order to derive an expression for the isospin density $\bar{n}_I$ and to confirm the usual thermodynamic relation between the free energy and the baryon number density $\bar{n}_B$. Similarly, the derivatives with respect to the source $q_3$ are expected to yield the spatial components of the baryon current. We have also included the angle $\alpha$ to verify as a consistency check that the free energy does not depend on this angle. From Eq. (40) we have

$$
\frac{\partial \bar{\Omega}}{\partial x} = \frac{1}{2} \int_{-\infty}^{\infty} dz \left[ \partial_z \left( \frac{\partial \mathcal{L}}{\partial A'} \frac{\partial A}{\partial x} \right) + \partial_z \left( \frac{\partial \mathcal{L}}{\partial \hat{A}'_i} \frac{\partial \hat{A}_i}{\partial x} \right) + \partial_z \left( \frac{\partial \mathcal{L}}{\partial K'_a} \frac{\partial K_a}{\partial x} \right) \right.
$$
$$
\left. + \partial_z \left( \frac{\partial \mathcal{L}}{\partial h'} \frac{\partial h}{\partial x} \right) + \partial_z \left( \frac{\partial \mathcal{L}}{\partial h'_3} \frac{\partial h_3}{\partial x} \right) \right] + \frac{2\bar{m}_\pi^2 \sin\theta}{9\pi} \frac{\partial \theta}{\partial x} . \tag{67}
$$

The various terms on the right-hand side can receive contributions from the ultraviolet boundary but also from the infrared discontinuities of $h$ and $h_3$ at $z = 0$ ($u = u_{\rm KK}$). The boundary condition (58) dictates that $A(z)$ is even in $z$, and its behavior near $u = u_{\rm KK}$ is

$$
A(u) = A_c + \mathcal{O}(u - u_{\rm KK}), \tag{68}
$$

with $A_c$ to be determined dynamically. Due to the boundary conditions (45), $K_1(z)$ and $K_2(z)$ are odd, while $K_3(z)$ is even, and their infrared behavior is

$$
K_1(u) = K_{1(1)} \sqrt{u - u_{\rm KK}} + \ldots = \frac{K_{1(1)}}{\sqrt{3 u_{\rm KK}}} z + \ldots, \tag{69a}
$$

$$
K_2(u) = K_{2(1)} \sqrt{u - u_{\rm KK}} + \ldots = \frac{K_{2(1)}}{\sqrt{3 u_{\rm KK}}} z + \ldots, \tag{69b}
$$

$$
K_3(u) = K_{3c} + \mathcal{O}(u - u_{\rm KK}), \tag{69c}
$$

with the slopes $K_{1(1)}$ and $K_{2(1)}$ and the boundary value $K_{3c}$ to be determined dynamically. The infrared behavior of the functions $h$ and $h_3$ is

$$
h(u) = h_c + h_{(1)} \sqrt{u - u_{\rm KK}} + \ldots = h_c + \frac{h_{(1)}}{\sqrt{3 u_{\rm KK}}} z + \ldots, \tag{70a}
$$

$$
h_3(u) = h_{3c} + h_{3(1)} \sqrt{u - u_{\rm KK}} + \ldots = h_{3c} + \frac{h_{3(1)}}{\sqrt{3 u_{\rm KK}}} z + \ldots \tag{70b}
$$

Here, $h_c, h_{3c}, h_{(1)}, h_{3(1)}$ are all a priori unknown. The parity of the functions $\hat{A}_i(z)$ can be read off of (64). For $h(z)$ and $h_3(z)$ being both odd, we find that $\hat{A}_1(z)$ and $\hat{A}_2(z)$ are even, while $\hat{A}_3(z)$ is odd, as already anticipated in the boundary conditions (59).

We can now go back to Eq. (67) to compute

$$
\frac{\partial \bar{\Omega}}{\partial x} = -\bar{n}_B \frac{\partial \bar{\mu}_B}{\partial x} + \frac{1}{2} \frac{\partial \bar{\mu}_I}{\partial x} (\kappa_1 \sin\theta \sin\alpha - \kappa_2 \sin\theta \cos\alpha - \kappa_3 \cos\theta) + \frac{2\bar{m}_\pi^2 \sin\theta}{9\pi} \frac{\partial \theta}{\partial x}
$$
$$
+ \frac{\bar{\mu}_I}{2} \frac{\partial \theta}{\partial x} (\kappa_1 \cos\theta \sin\alpha - \kappa_2 \cos\theta \cos\alpha + \kappa_3 \sin\theta) + \frac{\bar{\mu}_I \sin\theta}{2} \frac{\partial \alpha}{\partial x} (\kappa_1 \cos\alpha + \kappa_2 \sin\alpha)
$$
$$
+ \frac{1}{4} \frac{\partial h_c}{\partial x} \left( -\sqrt{3} u_{\rm KK}^2 h_{(1)} + 6\lambda_0 A_c h_c h_{3c} \right) + \frac{1}{8} \frac{\partial h_{3c}}{\partial x} \left( -\sqrt{3} u_{\rm KK}^2 h_{3(1)} + 6\lambda_0 A_c h_c^2 \right), \tag{71}
$$

where the third line comes from the discontinuities. We have abbreviated

$$\kappa_i \equiv (u^{5/2}\sqrt{f}K_i')_{u=\infty}, \tag{72}$$

which, using the equations of motion (65) and the expansions (69), can be expressed as

$$\kappa_1 = \frac{\sqrt{3}u_{\mathrm{KK}}^2}{2}K_{1(1)} + \lambda_0^2 \int_{u_{\mathrm{KK}}}^{\infty} du\, \frac{K_1\left[4u^2(h^2+h_3^2)+9h^2h_3^2\right]}{4u^{5/2}\sqrt{f}}, \tag{73a}$$

$$\kappa_2 = \frac{\sqrt{3}u_{\mathrm{KK}}^2}{2}K_{2(1)} + \lambda_0^2 \int_{u_{\mathrm{KK}}}^{\infty} du\, \frac{K_2\left[4u^2(h^2+h_3^2)+9h^2h_3^2\right]}{4u^{5/2}\sqrt{f}}, \tag{73b}$$

$$\kappa_3 = \lambda_0^2 \int_{u_{\mathrm{KK}}}^{\infty} du\, \frac{K_3 h^2(8u^2+9h^2)}{4u^{5/2}\sqrt{f}}. \tag{73c}$$

From Eq. (71) we can easily read off the derivatives with respect to all possible values of $x$. First of all, we see that the derivative with respect to $x = q_3$ vanishes. This is expected since we chose the corresponding integration constant in Eq. (64) such that $(u^{5/2}\sqrt{f}\hat{A}_3')_{z=\pm\infty} = 0$, enforcing a vanishing baryon current. This is equivalent to treating $q_3$ as a dynamical variable with respect to which we minimize the free energy. One can compute $q_3$ from

$$q_3 = \int_{u_{\mathrm{KK}}}^{\infty} du\, \hat{A}_3' = -\frac{3\lambda_0^2}{2}\int_{u_{\mathrm{KK}}}^{\infty} du\, \frac{h^2 K_3}{u^{5/2}\sqrt{f}}, \tag{74}$$

using that $\hat{A}_3(u = u_{\mathrm{KK}}) = 0$ since $\hat{A}_3(z)$ is odd and continuous.

For $x = \alpha$ we expect the derivative to vanish as well due to the symmetry of the system. We see that it vanishes trivially in the absence of a charged pion condensate, $\theta = 0$. If $\theta$ is non-zero, we use the following observation to show that the derivative is still zero. With partial integration and using the equation of motion for $K_1$ (65a) we have

$$\int_{u_{\mathrm{KK}}}^{\infty} du\, u^{5/2}\sqrt{f}K_1'K_2' = \kappa_1\frac{\bar{\mu}_I}{2}\sin\theta\cos\alpha - \lambda_0^2\int_{u_{\mathrm{KK}}}^{\infty} du\, \frac{K_1 K_2\left[4u^2(h^2+h_3^2)+9h^2h_3^2\right]}{4u^{5/2}\sqrt{f}}, \tag{75}$$

and, on the other hand, by exchanging the roles of $K_1$ and $K_2$ in the partial integration we get the alternative expression for the same integral

$$\int_{u_{\mathrm{KK}}}^{\infty} du\, u^{5/2}\sqrt{f}K_1'K_2' = -\kappa_2\frac{\bar{\mu}_I}{2}\sin\theta\sin\alpha - \lambda_0^2\int_{u_{\mathrm{KK}}}^{\infty} du\, \frac{K_1 K_2\left[4u^2(h^2+h_3^2)+9h^2h_3^2\right]}{4u^{5/2}\sqrt{f}}. \tag{76}$$

Subtracting Eq. (76) from Eq. (75) gives $\kappa_1\cos\alpha + \kappa_2\sin\alpha = 0$, and thus the free energy does indeed not depend on $\alpha$.

Next, we turn to the thermodynamic relations for baryon and isospin densities by choosing $x \in \{\bar{\mu}_B, \bar{\mu}_I\}$. For $x = \bar{\mu}_B$ we obtain

$$\bar{n}_B = -\frac{\partial\bar{\Omega}}{\partial\bar{\mu}_B}, \tag{77}$$

which is a consistency check since it is the thermodynamic relation we expect (but gives no further information). For $x = \bar{\mu}_I$, we obtain an expression for the isospin density,

$$\bar{n}_I = -\frac{\partial\bar{\Omega}}{\partial\bar{\mu}_I} = \frac{1}{2}\left[-(\kappa_1\sin\alpha - \kappa_2\cos\alpha)\sin\theta + \kappa_3\cos\theta\right], \tag{78}$$

which we shall employ below in our explicit calculation.

It remains to write down the conditions obtained from minimizing the free energy with respect to the parameters $\theta, h_c, h_{3c}$. Setting $x = \theta$ and requiring the derivative of $\bar{\Omega}$ with respect to $\theta$ to vanish, we read off of Eq. (71)

$$0 = \frac{1}{2}\Big[(\kappa_1 \sin \alpha - \kappa_2 \cos \alpha) \cos \theta + \kappa_3 \sin \theta\Big] + \frac{2\bar{m}_\pi^2}{9\pi\bar{\mu}_I} \sin \theta \,. \tag{79}$$

From the stationarity of the free energy with respect to $x \in \{h_c, h_{3c}\}$ we obtain two conditions which, assuming $h_c, h_{3c} \neq 0$, can be written as

$$A_c = \frac{u_{KK}^2 h_{(1)}}{2\sqrt{3}\lambda_0 h_c h_{3c}} \,, \tag{80a}$$

$$h_c h_{(1)} = h_{3c} h_{3(1)} \,. \tag{80b}$$

The three conditions (79), (80a), (80b) supplement the equations of motion (65), (66) and have to be solved simultaneously with them. We will explain our numerical procedure to do so at the beginning of Sec. 5.

Finally, we use the results just derived to write $\bar{\mu}_B$ and $\bar{\Omega}$ in a convenient form for the numerical evaluation. From Eqs. (62) and (80a) we have

$$\bar{\mu}_B = A_c + \int_{u_{KK}}^\infty du \, A'(u) = \frac{u_{KK}^2 h_{(1)}}{2\sqrt{3}\lambda_0 h_c h_{3c}} + \int_{u_{KK}}^\infty du \, \frac{\bar{n}_B Q}{u^{5/2}\sqrt{f}} \,. \tag{81}$$

For the free energy density, we employ partial integration and the equations of motion to first compute

$$\int_{u_{KK}}^\infty du \, \frac{u^{5/2}}{2\sqrt{f}} \left( f K_a' K_a' + g_3 \right) = \frac{\bar{\mu}_I \bar{n}_I}{2} + \frac{3}{4} \int_{u_{KK}}^\infty du \, h \left[ h_3(\hat{A}_1' K_1 + \hat{A}_2' K_2) + \hat{A}_3' h K_3 \right] \,, \tag{82a}$$

$$\int_{u_{KK}}^\infty du \, \frac{u^{5/2}\sqrt{f}}{2\lambda_0^2} \hat{A}_i' \hat{A}_i' = \frac{3}{4} \int_{u_{KK}}^\infty du \left[ \hat{A}_1(h h_3 K_1)' + \hat{A}_2(h h_3 K_2)' + \hat{A}_3(h^2 K_3)' \right] \,, \tag{82b}$$

$$\frac{3\lambda_0}{4} \int_{u_{KK}}^\infty du \, A(h^2 h_3)' = \bar{\mu}_B \bar{n}_B - \int_{u_{KK}}^\infty du \, \frac{(\bar{n}_B Q)^2}{u^{5/2}\sqrt{f}} \,. \tag{82c}$$

Inserting this into Eq. (40) with the Lagrangian (51) we get

$$\bar{\Omega} = \int_{u_{KK}}^\infty du \, \frac{u^{5/2}}{2\sqrt{f}} \left[ g_1 + g_2 + \frac{(\bar{n}_B Q)^2}{u^5} \right] - \bar{\mu}_B \bar{n}_B - \frac{\bar{\mu}_I \bar{n}_I}{2} - \frac{2\bar{m}_\pi^2}{9\pi}(\cos \theta - 1) \,. \tag{83}$$

This free energy is structurally identical to the one derived in Ref. [39], except for the additional mass term.

# 4 Different phases

There are various different configurations that solve our equations of motion, corresponding to different thermodynamic phases. These are summarized in Table 1. In the following we define these phases and discuss their physical content.

Table 1: Summary of the phases discussed in Sec. 4 in terms of the spatial and temporal non-abelian gauge components $h_{1,2}$, $h_3$ and $K_{1,2}$, $K_3$, as well as charged pion condensate $\theta$, baryon density $n_B$ and isospin density $n_I$. A "×" means the corresponding quantity is non-zero (and non-constant in $z$ in the case of the gauge components). The pion condensate is non-zero if and only if $K_{1,2}$ are non-zero. The baryon number $n_B$ is non-zero if and only if $h_{1,2}$ and $h_3$ are non-trivial. Isospin number $n_I$ sits in the asymptotic behaviors of $K_{1,2}$ and $K_3$. If $K_3 = $ const only $K_{1,2}$ contribute to $n_I$. Rho meson condensation is encoded in the functions $h$ and $h_3$ and is included without symmetry distinction in the baryonic phases. All phases listed here appear in the phase diagram, except for the $\rho$ phase, which only appears if pion condensation is ignored (or for unphysical model parameters). A second $\pi\rho$ phase where $h_3 = 0$ and $h \neq 0$ does exist as a solution but turns out to be irrelevant for the phase structure and is not discussed in the text.

|  | $h_{1,2}(z)$ | $h_3(z)$ | $K_{1,2}(z)$ | $K_3(z)$ | $\theta$ | $n_B$ | $n_I$ |
|---|---|---|---|---|---|---|---|
| V | 0 | 0 | 0 | const | 0 | 0 | 0 |
| $\pi$ | 0 | 0 | × | const | × | 0 | × |
| $\pi\rho$ | 0 | × | × | const | × | 0 | × |
| $\rho$ | × | 0 | 0 | const | 0 | 0 | × |
| B | × | × | 0 | × | 0 | × | × |
| $\pi$B | × | × | × | × | × | × | × |

## 4.1 Vacuum (V) and pion-condensed phase ($\pi$)

In the vacuum, baryon and isospin densities vanish. The pion condensate is zero, $\theta = 0$, and all relevant functions are constants,

$$h(u) = h_3(u) = K_1(u) = K_2(u) = \hat{A}_i(u) = 0, \quad A(u) = \bar{\mu}_B, \quad K_3(u) = \frac{\bar{\mu}_I}{2}. \tag{84}$$

The free energy has been normalized such that it vanishes in this case, $\bar{\Omega} = 0$.

The pion-condensed phase was already discussed in Sec. 2.3. It is recovered as a solution of our more general setup for

$$h(u) = h_3(u) = \hat{A}_i(u) = 0, \qquad A(u) = \bar{\mu}_B, \tag{85}$$

while the functions $K_a(u)$ are given by Eq. (46). The pion condensate, free energy, and isospin density are given in Eq. (48). In the chiral limit, $m_\pi = 0$, this phase reduces to the pion-condensed configuration considered in Ref. [39]. In the given setup (confined geometry and maximally separated flavor branes) the holographic $\pi$ phase is identical to what is obtained from lowest-order chiral perturbation theory.

## 4.2 Coexisting pion and rho meson condensates ($\pi\rho$) and rho meson phase ($\rho$)

Our motivation to introduce non-zero spatial components of the non-abelian gauge fields $h$ and $h_3$ was to describe baryonic matter. We now demonstrate that due to the difference in $h$ and $h_3$ our approach also enables us to include the physics of rho mesons, thus connecting our work with the analysis performed originally in Ref. [46].

Consider a configuration where $h(u) = 0$, but $h_3(u)$ is non-zero. On account of Eq. (56) this configuration has zero baryon number. Assuming a non-zero pion condensate, we simplify the calculation of the gauge potentials $K_a(u)$ by defining

$$\tilde{K}_1(u) = -\frac{2K_1(u)}{\bar{\mu}_I \sin\theta \sin\alpha}, \qquad \tilde{K}_2(u) = \frac{2K_2(u)}{\bar{\mu}_I \sin\theta \cos\alpha}, \qquad \tilde{K}_3(u) = \frac{2K_3(u)}{\bar{\mu}_I \cos\theta}. \tag{86}$$

The advantage of these new fields is that their boundary values are constant, irrespective of the (unknown) value of the pion condensate, $\tilde{K}_1(\infty) = \tilde{K}_2(\infty) = \tilde{K}_3(\infty) = 1$, which simplifies the numerical evaluation. The equations of motion (65a), (65b) and the boundary conditions (45a), (45b) are then the same for $\tilde{K}_1$ and $\tilde{K}_2$, and we can simply set

$$\tilde{K}_1(u) = \tilde{K}_2(u) \equiv \tilde{K}(u), \tag{87}$$

which corresponds to $K_1(u)\cos\alpha + K_2(u)\sin\alpha = 0$. Since $h = 0$, Eq. (65c) yields the same $K_3(u)$ as in the $\pi$ phase, which implies

$$\tilde{K}_3(u) = 1, \tag{88}$$

and it remains to solve (numerically) the system

$$(u^{5/2}\sqrt{f}\tilde{K}')' = \frac{\lambda_0^2 \tilde{K} h_3^2}{u^{1/2}\sqrt{f}}, \tag{89a}$$

$$(u^{5/2}\sqrt{f}h_3')' = -\frac{\bar{\mu}_I^2 \sin^2\theta\, \lambda_0^2 \tilde{K}^2 h_3}{u^{1/2}\sqrt{f}}, \tag{89b}$$

for $\tilde{K}$ and $h_3$. Due to $h(u) = 0$ we have $h_c = h_{(1)} = 0$. Hence, from Eq. (71) we see that stationarity with respect to $h_c$ is automatically fulfilled, while stationarity with respect to $h_{3c}$ yields

$$h_{3(1)} = 0, \tag{90}$$

i.e., the slope of the function $h_3(z)$ vanishes at $z = 0$.

The isospin density is computed from Eq. (78) with the help of Eqs. (73) and (86), and can be written as

$$\bar{n}_I = \bar{\mu}_I X \sin^2\theta, \tag{91}$$

with the abbreviation

$$X \equiv \frac{\sqrt{3}u_{\text{KK}}^2}{8}\tilde{K}_{(1)} + \frac{\lambda_0^2}{4}\int_{u_{\text{KK}}}^{\infty} du\, \frac{\tilde{K}h_3^2}{u^{1/2}\sqrt{f}}. \tag{92}$$

This quantity also appears in the stationarity condition with respect to $\theta$ (79), which becomes

$$\cos\theta = \frac{\bar{m}_\pi^2}{\bar{\mu}_I^2}\frac{2}{9\pi X}. \tag{93}$$

In the $\pi$ phase, the isospin density receives a contribution only from the first term in $X$. Now, in the presence of $h_3$, there is a second contribution. Also, the difference in the spatial components $A_3 \sim h_3 \neq 0$ and $A_{1,2} \sim h = 0$ renders the system anisotropic. We interpret this phase as rho meson condensation on top of the pion condensate and refer to it as the $\pi\rho$ phase. Indeed, Eqs. (89) are identical to Eqs. (4.21) in Ref. [46], where the rho meson condensate was sitting in the $\tau_{1,2}$ sector, as expected from a charged rho meson condensate in the usual frame. Here, due to our rotation (12) a $\tau_3$ component of the charged rho meson is generated, which is accounted for by $h_3$. We can further confirm the equivalence to the results of Ref. [46] by discussing the continuous connection to the $\pi$ phase. It is obvious from Eqs. (89a) and (93) that in the $h_3 \to 0$ limit $\tilde{K}$ and the pion condensate reduce to their solution in the pure $\pi$-phase,

$$\tilde{K}(z) = \frac{2}{\pi}\arctan\frac{z}{u_{\text{KK}}}, \qquad \cos\theta = \frac{\bar{m}_\pi^2}{\bar{\mu}_I^2}. \tag{94}$$

Now, if we are in the $\pi\rho$ phase, $h_3$ is non-zero and approaches zero as we approach the transition point to the $\pi$ phase. To find this point we employ Eq. (89b). We consider $\tilde{h}_3 = h_3/h_{3c}$ in order to work with a non-zero function with fixed boundary value $\tilde{h}_3(z = 0) = 1$. We also have $\tilde{h}'_3(z = 0) = 0$ due to Eq. (90), which holds for arbitrarily small values of $h_{3c}$. Also defining $\tilde{z} = z/u_{KK}$ and using Eq. (94), we can write Eq. (89b) close to the transition point – but within the $\pi\rho$ phase – as

$$\tilde{h}''_3 + \frac{2\tilde{z}}{1+\tilde{z}^2}\tilde{h}'_3 = -\frac{16\Lambda\tilde{h}_3 \arctan^2 \tilde{z}}{9\pi^2 u_{KK}(1+\tilde{z}^2)^{4/3}}\,, \tag{95}$$

where prime is now the derivative with respect to $\tilde{z}$, and where

$$\Lambda \equiv \lambda_0^2\bar{\mu}_I^2\left(1 - \frac{\bar{m}_\pi^4}{\bar{\mu}_I^4}\right). \tag{96}$$

For the given boundary conditions, Eq. (95) gives a tower of eigenvalues $\Lambda \simeq 1.90176$, $6.48327, 13.8360, 23.9705, \ldots$. For fixed model parameters $\lambda$, $\bar{m}_\pi$, these values translate into a critical chemical potential for the onset of a particular vector meson mode within the background of the pion condensate. We will only consider isospin chemical potentials large enough for the first, $\Lambda_1 = 1.90176$, but not for any higher mode to appear. In any case, at chemical potentials which allow for more than one vector meson to condense one would have to take into account the possibility of the coexistence of several condensates, while Eq. (96) only indicates the critical point for condensation of a single mode in the pionic background. Therefore, the relevant chemical potential is

$$\rho \text{ onset in pion condensate:} \qquad \bar{\mu}_I = \sqrt{\frac{\Lambda_1 + \sqrt{\Lambda_1^2 + 4\lambda_0^4\bar{m}_\pi^4}}{2\lambda_0^2}} \simeq \frac{\sqrt{\Lambda_1}}{\lambda_0}\,, \tag{97}$$

where the approximation holds for $m_\pi^2 \ll \Lambda M_{KK}^2/2$. Since in our physical results we will choose $M_{KK}$ to be of the order of $1\,\text{GeV}$, this is a good approximation.

The critical chemical potential (97) can be interpreted as the value of the effective rho meson mass (at that particular $\bar{\mu}_I$) in the pionic medium. It is therefore useful for comparison to also derive the vacuum mass of the rho meson within our setup. To this end, we switch off the pion condensate, $\theta = 0$, which allows us to set $K_1(u) = K_2(u) = 0$ due to the boundary conditions (45). Now, since in the absence of a pion condensate the rotation (12) is trivial, the rho meson condensate *does* sit in the $\tau_{1,2}$ sector. Therefore, we set $h_3 = 0$ and work with a non-zero $h$. The only non-trivial equations of motion are then Eqs. (65c) and (66a), which have to be solved numerically for $K_3$ and $h$. For our present purpose, we are again interested in the onset of rho meson condensation, this time in the vacuum, where $h = 0$. To this end, we can simply set $h = 0$ on the right-hand side of Eq. (65c) and find $\tilde{K}_3(u) = 1$. For Eq. (66a) we define $\tilde{h} = h/h_c$ to work with a non-zero and constant boundary value $\tilde{h}(z = 0) = 1$ to write this equation as

$$\tilde{h}'' + \frac{2\tilde{z}}{1+\tilde{z}^2}\tilde{h}' = -\frac{4\Lambda^{(0)}\tilde{h}}{9u_{KK}(1+\tilde{z}^2)^{4/3}}\,, \tag{98}$$

with $\Lambda^{(0)} \equiv \lambda_0^2\bar{\mu}_I^2$. Once again, we compute the eigenvalues numerically, $\Lambda^{(0)} \simeq 0.669314$, $2.87432, 6.59117, 11.7967, \ldots$. Equation (98) is identical to Eq. (4.4) in the original work by Sakai and Sugimoto [7], which describes vector and axial vector mesons in the WSS model (our boundary conditions select the vector mesons). The lowest mode, $\Lambda_1^{(0)} = 0.669314$, corresponds to the $\rho$ meson, and the corresponding critical chemical potential can be interpreted as the vacuum mass of the rho meson,

$$\bar{m}_\rho^2 = \frac{\Lambda_1^{(0)}}{\lambda_0^2}\,, \qquad m_\rho = \lambda_0 M_{KK}\bar{m}_\rho\,, \tag{99}$$

where we have introduced the same notation for the dimensionless and dimensionful versions of the $\rho$ mass as for the pion mass. We can thus express the critical chemical potential for the onset of rho meson condensation in the pionic background (97) in terms of the vacuum mass,

$$\bar{\mu}_I = \left[\sqrt{\frac{\Lambda_1}{\Lambda_1^{(0)}}} + \mathcal{O}\left(\frac{m_\pi^4}{m_\rho^4}\right)\right]\bar{m}_\rho \simeq 1.68\,\bar{m}_\rho\,. \tag{100}$$

Hence, the model predicts that the pionic medium increases the rho meson mass. The numerical factor is in agreement with Ref. [46], where the chiral limit was considered.

## 4.3 Baryonic matter (B)

Next, we consider the baryonic phase without pion condensate. We expect this phase to be relevant for baryon chemical potentials much larger than the isospin chemical potential. Since $\sin\theta = 0$ in this case, we can set $K_1(u) = K_2(u) = 0$. Consequently, $\hat{A}_1(u) = \hat{A}_2(u) = 0$. On the other hand, $h$ and $h_3$ must both be non-vanishing to generate baryon number. Hence, in the B phase we need to solve the following equations of motion,

$$(u^{5/2}\sqrt{f}\tilde{K}_3')' = \frac{2\lambda_0^2\tilde{K}_3 h^2}{u^{1/2}\sqrt{f}} + \frac{9\lambda_0^2\tilde{K}_3 h^4}{4u^{5/2}\sqrt{f}}\,, \tag{101a}$$

$$(u^{5/2}\sqrt{f}h')' = \frac{\lambda_0^2 h}{u^{1/2}\sqrt{f}}\left[h^2 + h_3^2 - \bar{\mu}_I^2\tilde{K}_3^2 - \frac{9}{4u^2}\left(h^2\bar{\mu}_I^2\tilde{K}_3^2 - \frac{4h_3\bar{n}_B Q}{3\lambda_0}\right)\right]\,, \tag{101b}$$

$$(u^{5/2}\sqrt{f}h_3')' = \frac{\lambda_0^2 h^2}{u^{1/2}\sqrt{f}}\left(2h_3 + \frac{3\bar{n}_B Q}{\lambda_0 u^2}\right)\,. \tag{101c}$$

The isospin density (78) assumes the form

$$\bar{n}_I = \bar{\mu}_I\lambda_0^2\int_{u_{\mathrm{KK}}}^{\infty} du\,\frac{\tilde{K}_3 h^2(8u^2 + 9h^2)}{16u^{5/2}\sqrt{f}}\,. \tag{102}$$

Any non-zero $\bar{\mu}_I$ produces *deformed* baryons with $h \neq h_3$, and the B phase can be continuously connected to the $\rho$ phase discussed in the previous subsection. In the $\rho$ phase, since $h_3 = 0$ and $h \neq 0$, we have $h_{3(1)} = 0$ while $h_c \neq 0$. These conditions, inserted into the stationarity condition for $h_{3c}$ from Eq. (71), yield $A_c = 0$. With $\bar{n}_B = 0$ and Eq. (81), this implies $\bar{\mu}_B = 0$. Therefore, the B phase can connect continuously to the pure $\rho$ phase only for vanishing baryon chemical potential,

$$\text{baryon onset in } \rho: \qquad \bar{\mu}_B = 0\,. \tag{103}$$

The B phase can also connect continuously to the vacuum. Letting both $h$ and $h_3$ go to zero in Eqs. (101), the only term on the right-hand sides that is not of higher order is the term proportional to $h\bar{\mu}_I^2\tilde{K}_3^2$ in Eq. (101c). One arrives at Eq. (98), which gives the critical isospin chemical potential $\bar{\mu}_I = \bar{m}_\rho$. Since $h$ and $h_3$ go to zero, both stationarity conditions for $h_c$ and $h_{3c}$ from Eq. (71) are trivially fulfilled and thus there is no condition for $A_c$. As a consequence, this second-order onset does not depend on $\bar{\mu}_B$ and is a vertical line in the $\mu_B$-$\mu_I$ phase diagram. In fact, as the full numerical calculation shows, this onset occurs in a metastable regime and is only realized in the phase diagram if pion condensation is omitted.

## 4.4 Baryonic matter coexisting with a pion condensate ($\pi$B)

This is the most complicated case since all functions are non-trivial and we have non-vanishing pion condensate, baryon density, and isospin density. We employ the redefined gauge potentials (86) and may set $\tilde{K}_1(u) = \tilde{K}_2(u) \equiv \tilde{K}(u)$ to solve the equations of motion (65) and (66), which now read

$$(u^{5/2}\sqrt{f}\tilde{K}')' = \frac{\lambda_0^2 \tilde{K}(h^2 + h_3^2)}{u^{1/2}\sqrt{f}} + \frac{9\lambda_0^2 \tilde{K}h^2 h_3^2}{4u^{5/2}\sqrt{f}}, \tag{104a}$$

$$(u^{5/2}\sqrt{f}\tilde{K}_3')' = \frac{2\lambda_0^2 \tilde{K}_3 h^2}{u^{1/2}\sqrt{f}} + \frac{9\lambda_0^2 \tilde{K}_3 h^4}{4u^{5/2}\sqrt{f}}, \tag{104b}$$

$$(u^{5/2}\sqrt{f}h')' = \frac{\lambda_0^2 h}{u^{1/2}\sqrt{f}}\left\{ h^2 + h_3^2 - \bar{\mu}_I^2\left(\frac{\tilde{K}^2}{2}\sin^2\theta + \tilde{K}_3^2 \cos^2\theta\right) \right.$$

$$\left. - \frac{9}{2u^2}\left[\frac{\bar{\mu}_I^2}{4}\left(h_3^2\tilde{K}^2\sin^2\theta + 2h^2\tilde{K}_3^2\cos^2\theta\right) - \frac{2h_3\bar{n}_B Q}{3\lambda_0}\right] \right\}, \tag{104c}$$

$$(u^{5/2}\sqrt{f}h_3')' = \frac{\lambda_0^2}{u^{1/2}\sqrt{f}}\left[ h_3(2h^2 - \bar{\mu}_I^2\tilde{K}^2\sin^2\theta) - \frac{9h^2}{4u^2}\left(\bar{\mu}_I^2 h_3\tilde{K}^2\sin^2\theta - \frac{4\bar{n}_B Q}{3\lambda_0}\right) \right]. \tag{104d}$$

Obviously, by the redefinition (86) the pion condensate appears in the equations of motion themselves rather than in the boundary conditions. The isospin density (78) can be written as

$$\bar{n}_I = \bar{\mu}_I\left[ X\sin^2\theta + \lambda_0^2\int_{u_{\text{KK}}}^{\infty} du\, \frac{\tilde{K}_3 h^2\left(8u^2 + 9h^2\right)}{16u^{5/2}\sqrt{f}} \right], \tag{105}$$

with

$$X \equiv \frac{\sqrt{3}u_{\text{KK}}^2}{8}\tilde{K}_{(1)} + \frac{\lambda_0^2}{4}\int_{u_{\text{KK}}}^{\infty} du\left[ \frac{\tilde{K}(h^2 + h_3^2) - 2\tilde{K}_3 h^2}{u^{1/2}\sqrt{f}} + \frac{9h^2(\tilde{K}h_3^2 - \tilde{K}_3 h^2)}{4u^{5/2}\sqrt{f}} \right]. \tag{106}$$

Expressed in terms of $X$, the stationarity condition of the free energy with respect to the pion condensate (79) becomes the same as in the $\pi\rho$ phase,

$$\cos\theta = \frac{\bar{m}_\pi^2}{\bar{\mu}_I^2}\frac{2}{9\pi X}. \tag{107}$$

Just like in the $\pi\rho$ phase, the pion condensate of the $\pi$ phase (and chiral perturbation theory), $\cos\theta = \bar{m}_\pi^2/\bar{\mu}_I^2$, is corrected by a medium dependent term. Here, this term includes the effect of baryons and of rho meson condensation. We see from Eq. (107) that in the chiral limit, $\bar{m}_\pi = 0$, the pion condensate is maximal, irrespective of the medium (we have checked numerically that $X$ remains non-zero in the chiral limit). In the presence of a pion mass, Eq. (107) yields an expression for the critical isospin chemical potential for the onset of a pion condensate within baryonic matter. Namely, upon setting $\cos\theta = 1$, thus assuming a second-order onset,

$$\text{pion onset in B:}\qquad \bar{\mu}_I^2 = \bar{m}_\pi^2\frac{2}{9\pi X}. \tag{108}$$

We shall indeed find that there is a region in the phase diagram where this second-order onset is realized. The specific value for the critical isospin chemical potential at a given baryon

chemical potential – and thus the medium-dependent pion mass – has to be computed numerically.

As the B phase connects continuously to the $\rho$ phase, the $\pi$B phase connects continuously to the $\pi\rho$ phase. To describe this transition, we need to take the limit $h(u) \to 0$. We can set $h = 0$ in Eqs. (104a), (104b), (104d), and these equations form a closed system for the functions $\tilde{K}$, $\tilde{K}_3$, and $h_3$, which are exactly the equations of the $\pi\rho$ phase discussed in Sec. 4.2. The onset of baryons is obtained with the help of Eq. (104c). We replace $h = h_c \tilde{h}$ and take the limit $h_c \to 0$ to obtain

$$(u^{5/2}\sqrt{f}\tilde{h}')' = \frac{\lambda_0^2 \tilde{h}}{u^{1/2}\sqrt{f}}\left[h_3^2 - \bar{\mu}_I^2\left(\frac{\tilde{K}^2}{2}\sin^2\theta + \cos^2\theta\right) - \frac{9\bar{\mu}_I^2}{8u^2}h_3^2\tilde{K}^2\sin^2\theta\right], \tag{109}$$

where we have used $\tilde{K}_3 = 1$. Inserting the (numerical) solutions for $\tilde{K}$ and $h_3$ from the $\pi\rho$ phase, this can be solved for $\tilde{h}(u)$ with boundary condition $\tilde{h}(z = 0) = 1$ [and $\tilde{h}(\infty) = 0$]. Now, the baryon chemical potential from Eq. (81) becomes, setting $\bar{n}_B = 0$,

$$\text{baryon onset in } \pi\rho: \qquad \bar{\mu}_B = \frac{u_{\mathrm{KK}}^2}{2\sqrt{3}\lambda_0}\frac{\tilde{h}_{(1)}}{h_{3c}}. \tag{110}$$

This is the critical chemical potential for a second-order onset of baryons within the $\pi\rho$ phase, where $h_{3c}$ is computed within the $\pi\rho$ phase and $\tilde{h}_{(1)}$ is determined from solving Eq. (109). In contrast to the baryon onset in the $\rho$ phase, see Eq. (103), the onset in the presence of a pion condensate (110) occurs at a non-zero (and $\bar{\mu}_I$ dependent) $\bar{\mu}_B$.

## 4.5 Neutron star matter

We may use our setup to address the question whether pion condensation takes place in neutron stars. To this end, we add a (non-interacting) lepton gas of electrons and muons, impose electric charge neutrality and equilibrium with respect to the electroweak interaction. Following Refs. [39, 40], this can be done in the B phase by assigning electric charges 0 and +1 to the two isospin components, i.e., by interpreting baryon and isospin densities to be composed of neutron and proton densities, although neutron and proton states are not explicitly present in our calculation. When meson condensates are added, it is no longer obvious in our setup how to assign the electric charges because baryonic and mesonic contributions arise from the same gauge fields in the bulk. However, it will turn out not to be necessary to include pions – let alone rho mesons – here: in contrast to Refs. [39, 40] we have included a non-zero pion mass and thus we can check whether (and will confirm that) neutron star conditions are fulfilled in the B phase in a region where there is no stable solution of the $\pi$B phase.

Due to the different convention for the isospin chemical potential compared to Refs. [39, 40] it is useful for clarity to briefly recapitulate the conditions of electroweak equilibrium and neutrality. We define (dimensionless) neutron and proton chemical potentials by

$$\bar{\mu}_n = \bar{\mu}_B + \frac{\bar{\mu}_I}{2}, \qquad \bar{\mu}_p = \bar{\mu}_B - \frac{\bar{\mu}_I}{2}, \tag{111}$$

and (dimensionless) neutron and proton densities by

$$\bar{n}_n = \frac{\bar{n}_B}{2} + \bar{n}_I, \qquad \bar{n}_p = \frac{\bar{n}_B}{2} - \bar{n}_I. \tag{112}$$

Electroweak equilibrium with respect to the processes $p + e \to n + \nu_e$, $n \to p + e + \bar{\nu}_e$ reads

$$\mu_e = N_c\lambda_0 M_{\mathrm{KK}}(\bar{\mu}_n - \bar{\mu}_p) = N_c\lambda_0 M_{\mathrm{KK}}\bar{\mu}_I, \tag{113}$$

where $\mu_e$ is the (dimensionful) electron chemical potential and we have neglected the neutrino chemical potential. We also assume equilibrium with respect to purely leptonic processes converting an electron into a muon, such that electron and muon chemical potentials are identical, $\mu_e = \mu_\mu$. Due to electric charge neutrality, the proton density must be the same as the lepton density, which we can write as

$$\frac{\bar{n}_B}{2} - \bar{n}_I = \frac{3\pi^2}{\lambda_0^2 M_{\text{KK}}^3}[n_e(\mu_e) + n_\mu(\mu_e)], \tag{114}$$

where the (dimensionful) lepton densities are

$$n_\ell(\mu_\ell) = \Theta(\mu_\ell - m_\ell)\frac{(\mu_\ell^2 - m_\ell^2)^{3/2}}{3\pi^2}, \tag{115}$$

with $\ell = e, \mu$, electron mass $m_e \simeq 511\,\text{keV}$, and muon mass $m_\mu \simeq 106\,\text{MeV}$. The condition (114) has to be solved simultaneously with the relevant equations of the B phase in Sec. 4.3. We perform this calculation to indicate the location of beta-equilibrated, charge neutral matter in the phase diagram (blue lines in Figs. 1 and 7); for all other results $\bar{\mu}_B$ and $\bar{\mu}_I$ are independent and the results of this subsection play no role.

# 5 Numerical evaluation

In this section we evaluate the phases introduced in the previous section. Except for the $\pi$ phase, whose analytical solutions were already discussed in Sec. 2.3, this has to be done numerically. We start by explaining our numerical procedure. For concreteness, we do so for the most complicated case, the $\pi$B phase, the other configurations are treated similarly and more easily. The main difference to Ref. [39] in the practical calculation is, besides the larger number of functions, the determination of the pion condensate. As in Ref. [39], the most straightforward calculation is to consider fixed $\bar{\mu}_I$ and $\bar{n}_B$ and compute $\bar{\mu}_B$ and $\bar{n}_I$ after solving the differential equations. The difference in these quantities is that $\bar{\mu}_I$ and $\bar{n}_B$ appear in the boundary conditions while $\bar{\mu}_B$ and $\bar{n}_I$ are given by the non-trivial expressions (81) and (105), which require the solution of the equations of motion. When we need to work at fixed $\bar{\mu}_B$ and/or $\bar{n}_I$ – for example to determine the preferred phase at a fixed point in the $\bar{\mu}_B$-$\bar{\mu}_I$ phase diagram – we need to add Eq. (81) and/or Eq. (105) to the equations of motion. Here, we explain the simpler case where $\bar{\mu}_I$ and $\bar{n}_B$ are fixed. In addition to fixing these thermodynamic quantities, we also need to fix the model parameters $\bar{m}_\pi$ and $\lambda$ (for the solution in terms of our dimensionless quantities, the third model parameter $M_{\text{KK}}$ does not have to be fixed). The specific choice of the model parameters is explained in Sec. 5.1.

We deal with the stationarity equation for the pion condensate by defining the auxiliary function

$$\xi(u) \equiv \int_{u_{\text{KK}}}^{u} dv \left[ \frac{\tilde{K}(h^2 + h_3^2) - 2\tilde{K}_3 h^2}{v^{1/2}\sqrt{f}} + \frac{9h^2(\tilde{K}h_3^2 - \tilde{K}_3 h^2)}{4v^{5/2}\sqrt{f}} \right]. \tag{116}$$

By definition, this function obeys the first-order differential equation

$$\xi'(u) = \frac{\tilde{K}(h^2 + h_3^2) - 2\tilde{K}_3 h^2}{u^{1/2}\sqrt{f}} + \frac{9h^2(\tilde{K}h_3^2 - \tilde{K}_3 h^2)}{4u^{5/2}\sqrt{f}}, \tag{117}$$

with boundary condition $\xi(u_{\text{KK}}) = 0$. The pion condensate (107) can then be written purely in terms of boundary values/derivatives because

$$X = \frac{\sqrt{3}u_{\text{KK}}^2}{8}\tilde{K}_{(1)} + \frac{\lambda_0^2}{4}\xi(\infty). \tag{118}$$

We need to solve a system of 5 coupled differential equations – the equations of motion (104) and Eq. (117) – for the 5 functions $\tilde{K}$, $\tilde{K}_3$, $h$, $h_3$, $\xi$. We found it numerically advantageous to work in the $z$ variable on one half of the connected flavor branes. More precisely, we use $\tilde{z} = z/u_{\text{KK}}$ with $\tilde{z} \in [0, \infty]$. The calculation can be set up as an initial value problem with initial conditions at $\tilde{z} = 0$ (prime denoting derivative with respect to $\tilde{z}$)

$$\tilde{K}(0) = 0, \quad \tilde{K}'(0) = p_1, \quad \tilde{K}_3(0) = p_2, \quad \tilde{K}_3'(0) = 0, \quad \xi(0) = 0,$$

$$h(0) = p_3, \quad h'(0) = p_4, \quad h_3(0) = -\frac{4\bar{n}_B}{3\lambda_0 p_3^2}, \quad h_3'(0) = -\frac{3\lambda_0}{4\bar{n}_B}p_3^3 p_4, \tag{119}$$

where we have introduced the variables $p_1, p_2, p_3, p_4$, and where we have expressed the initial conditions of $h_3$ in terms of $\bar{n}_B$, $h(0)$, and $h'(0)$ with the help of Eqs. (56) and (80b). Also, to make all unknowns explicit, we denote the boundary value $\xi(\infty)$ by $p_5$. We formulate the initial value problem in *Mathematica* with the help of *ParametricNDSolve* for the parameters $\bar{\mu}_I, \bar{n}_B, p_1, p_2, p_3, p_4, p_5$ and perform the actual calculation via *FindRoot*, solving the set of 5 equations

$$\tilde{K}(\infty) = 1, \quad \tilde{K}_3(\infty) = 1, \quad h(\infty) = 0, \quad h_3(\infty) = 0, \quad p_5 = \xi(\infty), \tag{120}$$

for the 5 variables $p_1, p_2, p_3, p_4, p_5$. This determines the solutions $\tilde{K}$, $\tilde{K}_3$, $h$, $h_3$, $\xi$, which can then straightforwardly be used to compute $\cos\theta$, $\bar{\mu}_B$, $\bar{n}_I$, and the free energy density $\bar{\Omega}$ (83).

This procedure can be employed for the thermodynamic properties of the various phases but also for the phase diagram in the $\bar{\mu}_B$-$\bar{\mu}_I$ plane. To this end, the phase with lowest free energy has to be determined for each pair $(\bar{\mu}_B, \bar{\mu}_I)$. It is very tedious to compute the free energy for all possible phases in a sufficiently fine grid in the $\bar{\mu}_B$-$\bar{\mu}_I$ plane. Therefore, we set up specific calculations for the first-order phase transition curves, which simultaneously solve for the two phases that coexist on such a line supplemented by the condition that their free energies be the same. The relevant second-order transitions can also be computed directly along the lines already discussed in the previous section for the specific phases. The resulting phase diagram – the main result of our paper – is shown in Fig. 1 and discussed in Sec. 1. It includes the curve in the $\mu_B$-$\mu_I$ plane for neutron star matter, where we have added the constraints of Sec. 4.5. For this phase diagram we have fitted our parameters to basic meson properties of QCD. This is explained in the following subsection, together with an alternative fit that takes into account properties of symmetric nuclear matter.

## 5.1 Parameter fit

The parameters of our model are the Kaluza-Klein scale $M_{\text{KK}}$, the 't Hooft coupling $\lambda$ and the pion mass, say in its dimensionless form $\bar{m}_\pi$. The simplest fit reproduces the vacuum masses of pion and rho and the pion decay constant. These conditions give $M_{\text{KK}}$ from the rho meson mass via Eq. (99), then $\lambda$ from the pion decay constant via (35), and finally $\bar{m}_\pi$ from the pion mass via Eq. (37), resulting in

$$\text{``mesonic fit'':} \qquad M_{\text{KK}} = 949\,\text{MeV}, \qquad \bar{m}_\pi = 0.112, \qquad \lambda = 16.6. \tag{121}$$

This is the fit used for the main result in Fig. 1.

This fit completely ignores the properties of nuclear matter, which indeed are not reproduced accurately, as discussed at the end of Sec. 1.2. We might – alternatively – take a more global view and anchor our results to the locations of the phase transitions on the two axes that are known from QCD. On the $\mu_B$ axis, there is a first-order onset of isospin-symmetric nuclear matter at a critical quark chemical potential $\mu_0/N_c \simeq 308\,\text{MeV}$. On the $\mu_I$ axis, there

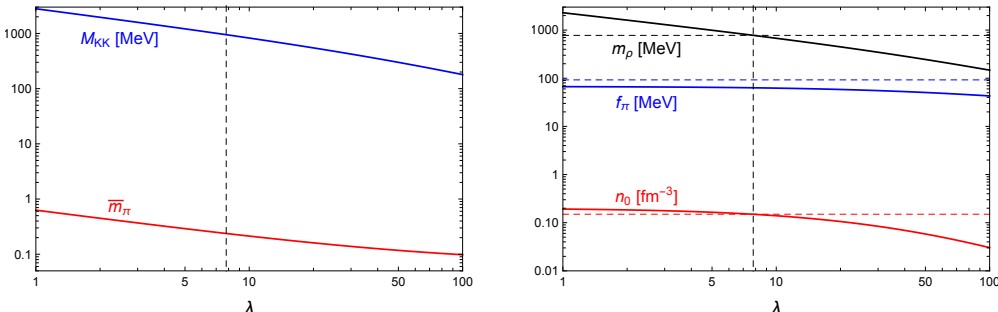

Figure 2: *Left panel:* Kaluza-Klein mass $M_{\text{KK}}$ and dimensionless pion mass $\bar{m}_\pi$ as functions of $\lambda$ given by the matching conditions (122). *Right panel:* Vacuum mass of the rho meson $m_\rho$, pion decay constant $f_\pi$, and saturation density of isospin-symmetric nuclear matter $n_0$ as functions of $\lambda$ resulting from $M_{\text{KK}}$ of the left panel. The horizontal dashed lines are the corresponding physical values. The vertical dashed line in both panels marks $\lambda = 7.8$, the value chosen for our "combined fit", which reproduces the correct $m_\rho$ and $n_0$, but leads to a deviation from the physical value of $f_\pi$.

is the second-order onset of pion condensation at $\mu_I = m_\pi \simeq 140\,\text{MeV}$. Requiring these values to be reproduced in our model amounts to setting

$$\lambda_0 N_c M_{\text{KK}} \bar{\mu}_B = \mu_0\,, \qquad \lambda_0 M_{\text{KK}} \bar{m}_\pi = m_\pi\,, \tag{122}$$

where $\bar{\mu}_B$ is the chemical potential at the baryon onset, which needs to be calculated numerically. Obviously, with our 3 parameters we cannot fulfill these constraints *and* reproduce the other basic quantities $n_0, f_\pi, m_\rho$. We proceed with the following observation. The dimensionless chemical potential $\bar{\mu}_B$ in Eq. (122) assumes different values for different values of $\lambda$, but does not depend on $M_{\text{KK}}$ [it also does not depend on $\bar{m}_\pi$ since there are no terms of the form $m_\pi n_B$ in our effective action, see discussion below Eqs. (52)] . As a consequence, Eq. (122) fixes $M_{\text{KK}}$ and $\bar{m}_\pi$ as functions of $\lambda$, shown in the left panel of Fig. 2. With these functions at hand, we can compute the baryon density at the first-order baryon onset $n_0$ (numerically), $f_\pi$ [from Eq. (35)], and $m_\rho$ [from Eq. (99)] as functions of $\lambda$, see right panel of Fig. 2.

We see that in this approach we have to live with an unphysical pion decay constant, no matter which $\lambda$ we choose. This tension between fitting vacuum properties and properties of nuclear matter simultaneously in the WSS model is known [36, 40]. We also see that, interestingly, by an appropriate choice of $\lambda$ we can fit *two* more quantities to good accuracy, namely $n_0$ and $m_\rho$. This motivates our alternative parameter choice:

$$\text{"combined fit":} \qquad M_{\text{KK}} = 949\,\text{MeV}\,, \qquad \bar{m}_\pi = 0.24\,, \qquad \lambda = 7.8\,. \tag{123}$$

The (unphysical) value of the pion decay constant for this choice is $f_\pi = 63.6\,\text{MeV}$. We discuss the resulting phase diagram together with the phase structure in the limit of large $\lambda$ in Sec. 5.5. All results in the following subsections 5.2 – 5.4 are produced with the "mesonic fit" (121) and thus they correspond to the phase diagram in Fig. 1.

## 5.2 Validity of the diagonal approximation

We have pointed out that our diagonal ansatz (50) is not a consistent solution of the full equations of motion, see text below this ansatz. Having set up our numerical evaluation, we are now in the position to test the validity of our approximation. To this end, we consider the phase diagram in the $\mu_B$-$\mu_I$ plane and employ the two limits mentioned below Eq. (50).

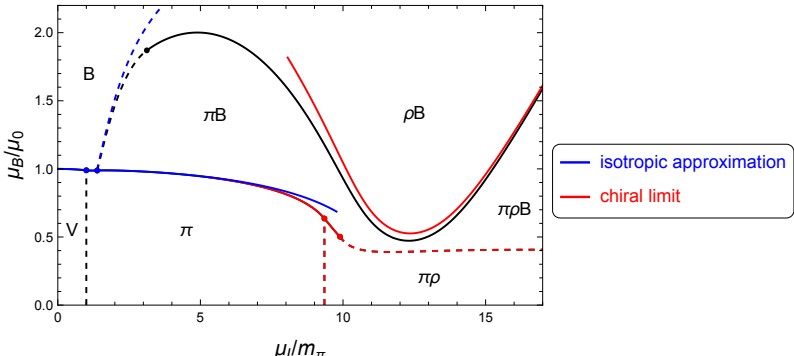

Figure 3: Comparison of our calculation within the diagonal ansatz (black curves, same as in Fig. 1) with the isotropic approximation (blue curves) and the chiral limit $m_\pi = 0$ (red curves). The black baryon onset curves (transitions from V, $\pi$, $\pi\rho$ to B and $\pi$B) are, for the naked eye, completely covered by one (or both) of these limits.

Firstly, we re-compute the phase transition lines at small $\mu_I$ in the isotropic approximation, which we know is a consistent solution to the equations of motion at $\mu_I = 0$. More precisely, we assume that the only non-zero spatial gauge components in the Lagrangian (28) are the diagonal components $h \equiv h_1 = h_2 = h_3$ and derive the equations of motion under this constraint, i.e., for the single function $h$ and the usual temporal components of the gauge fields. Of course, this approach does not yield a fully consistent solution for $\mu_I > 0$ either. The idea is rather to identify the regions in the phase diagram where our baryonic matter is approximately isotropic: in the regions where the isotropic approximation is in good agreement with our diagonal, but anisotropic, approach, we can expect that the full anisotropic approach does not differ much from the solution we have found. In Fig. 3 we show the isotropic approximation (blue curves) in comparison with our phase transition lines from Fig. 1 (black curves). We see that for not too large $\mu_I$ the two sets of lines are basically indistinguishable (the pion onset from the vacuum is irrelevant for this comparison since it does not involve any spatial components of the gauge fields). Expected deviations occur in the pion onset within baryonic matter as $\mu_B$ is increased and in the baryon onset within the pion-condensate phase as $\mu_I$ is increased.

The second limit we show in Fig. 3 is the chiral limit $m_\pi = 0$. In this limit, the full equations of motion *are* fulfilled without any off-diagonal components for arbitrary anisotropy in the diagonal components. This is best seen by employing the rotation of Refs. [39, 46, 58], i.e., $g_R = \mathbb{1}$, $g_L = \Sigma_0^\dagger$, instead of the rotation (12), which is more conveniently employed in the physical case. As explained below Eq. (12), the rotation $g_R = \mathbb{1}$, $g_L = \Sigma_0^\dagger$ does not generate $\tau_1$ and $\tau_2$ components if applied to the isospin chemical potential in the $\tau_3$ component. Therefore, two of the non-abelian temporal gauge fields can be set to zero, which is sufficient to also keep the off-diagonal spatial components switched off. Since we expect the chiral limit to be a good approximation for $\mu_I \gg m_\pi$, we have re-computed the phase transition lines in that regime setting $m_\pi = 0$ (red curves). For low baryon chemical potentials, we see that the curves are indistinguishable from our $m_\pi > 0$ results. One can check from the analytical expression for the critical chemical potential of the $\pi$-$\pi\rho$ transition (97) that the pion mass gives a correction smaller than 0.01 % to the location of this transition. Perhaps more surprisingly, the chiral limit also approximates the $\pi$-$\pi$B transition very well, essentially all the way down to the pion mass (where the red curve is covered by the blue curve), despite the pion condensate being far from maximal as $\mu_I \to m_\pi$. However, inclusion of the pion mass does result in a stronger deviation from the chiral limit in the $\pi$B-B transition.

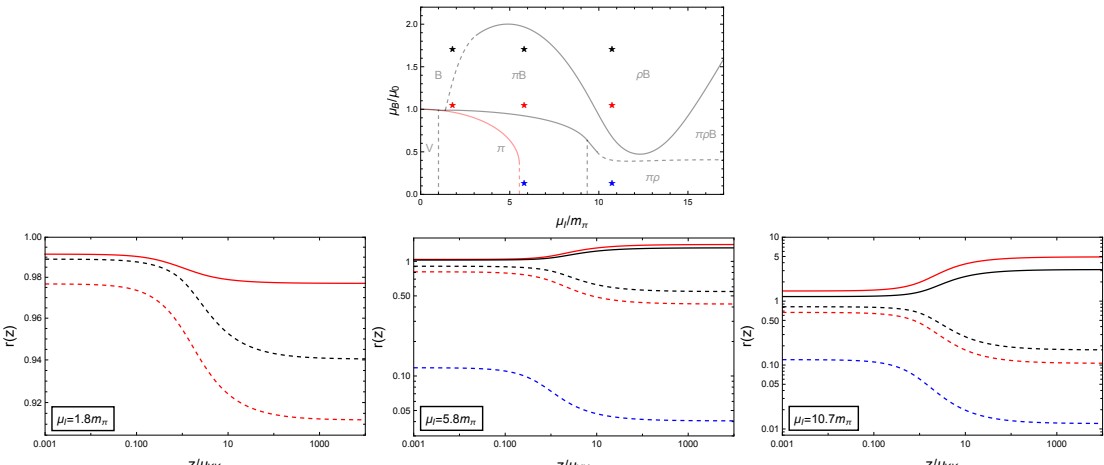

Figure 4: Ratio of the spatial components of the non-abelian gauge fields $r(z) = h_3(z)/h(z)$ for the $\pi$B phase (solid lines) and the B phase (dashed lines) for different values of $\mu_I$. In each panel, $\mu_B/\mu_0 = 0.13$ (blue), 1.05 (red), 1.71 (black). All points $(\mu_I, \mu_B)$ are indicated in the phase diagram, which is reproduced from Fig. 1. Note the difference in vertical scales in the three panels; the anisotropy [deviation of $r(z)$ from 1] increases strongly from left to right, interpreted as a larger admixture of a rho meson condensate.

Taking the results of the two limits together, the main conclusion from Fig. 3 is as follows. The agreement of our diagonal, anisotropic, non-zero $m_\pi$ approximation with the red and blue curves gives us confidence that our approximation works well in sizable regions of the phase diagram at small and large $\mu_I$ and even for all $\mu_I$ if $\mu_B$ is not too large. Our ansatz provides an interpolation between these two limits which is strictly speaking uncontrolled since we have dropped the off-diagonal components of the gauge fields. However, Fig. 3 suggests that – given the isotropic and chiral limits – there is not much room left for any additional structure other than what our ansatz yields. Therefore, the qualitative shape of our phase diagram seems to be a robust prediction.

## 5.3 Anisotropy in the baryonic phases

We have just seen with the help of the phase transition lines that the baryonic phases (B and $\pi$B) are isotropic to a good approximation for small $\mu_I$. Due to rho meson condensation the anisotropy will become more significant for large $\mu_I$. In Fig. 4, we show the extent of the anisotropy quantitatively. To this end, we have plotted the ratio $r(z) \equiv h_3(z)/h(z)$ for the B and $\pi$B phases at various points $(\mu_B, \mu_I)$, which, to facilitate the interpretation, we have marked in the phase diagram reproduced above the plots. For two of the points, there is no solution for the $\pi$B configuration (no blue solid lines in the middle and right panels), while solutions for both configurations exist at all other selected points, although, of course, at least one of the two phases is only metastable for a given point. We have also omitted the unstable $\pi$B solution that exists for the upper left point (no black solid line in the left panel). The first general observation is that the numerical value of the ratio $r(z)$ differs significantly between the infrared and ultraviolet regimes, $z \to 0$ and $z \to \infty$. For the following interpretation we may either focus on the infrared or on the ultraviolet, the main conclusions are the same.

For both B and $\pi$B phases, we see that the anisotropy is relatively small for the smallest isospin chemical potential used here, as expected from the comparison with the isotropic approximation in the previous subsection. Note that this "small" $\mu_I$ is already larger than the maximal $\mu_I$ found in neutron stars (constructed from our model). As expected, the anisotropy

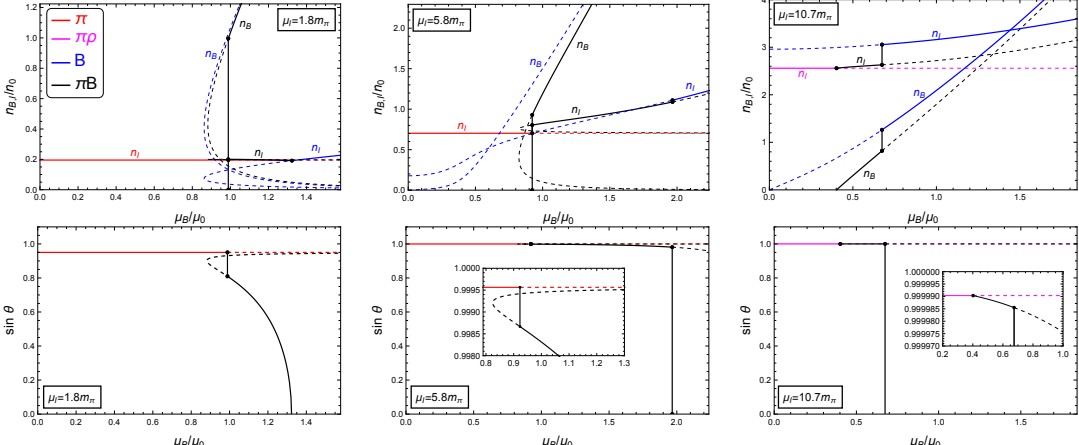

Figure 5: *Upper panels:* baryon and isospin densities in units of the saturation density $n_0$ as a function of the baryon chemical potential for three different isospin chemical potentials. In each panel, all phases that are favored in at least one domain of the panel are shown. When they are favored, the corresponding curve is solid; unstable and metastable phases are shown as dashed curves. Different phases are distinguished by color, as given in the legend in the upper left panel. Discontinuities due to first-order phase transitions are marked with vertical (black) lines. *Lower panels:* Pion condensate for the same isospin chemical potentials as in the upper panels. In regions where the B phase is favored, the pion condensate is zero.

tends to get larger as $\mu_I$ is increased at fixed $\mu_B$ or as $\mu_B$ is decreased at fixed $\mu_I$. The purely baryonic phase, i.e., the B phase, has values of $r(z)$ which are smaller than 1 everywhere. The reason is that this phase "wants" to turn into the $\rho$ phase, where $h_3 = 0$ and the rho meson condensate sits in the first two components in flavor space, i.e., in $h$. In contrast, in the extreme anisotropic limit the $\pi$B phase "wants" to turn into the $\pi\rho$ phase, where $h = 0$ and the $\rho$ meson condensate sits in $h_3$. [For the location of this transition see Eq. (110).] Hence we expect $r(z) > 1$, which is indeed the case except for a small region close to the B-$\pi$B transition (see red solid line in the left panel).

It might be tempting to translate these results into phase transition lines marking the onset of rho meson condensation within each of the baryonic phases. However, the results show that these transitions are not very sharp (at least for the physical value of the 't Hooft coupling chosen here). Perhaps one can argue that in the B phase the jump from the black to the red curves is much smaller than that of the red to the blue curve in both middle and right panels, and thus this transition is, roughly speaking, horizontal in the phase diagram. In the $\pi$B phase a similar argument (not immediately obvious from the curves shown here) suggests a more vertical transition. In any case, since in our approximation rho meson condensation in baryonic matter does not break any additional symmetries, there is no rigorous criterion for the transition and we do not attempt to add any phase transition lines to the phase diagram. We have, instead, included labels $\rho$B and $\pi\rho$B to indicate the approximate locations of these baryonic phases with a rho meson admixture.

## 5.4 Thermodynamic properties

We discuss the physical properties of the various phases by plotting the densities $n_B$, $n_I$ and the pion condensate $\sin\theta$ in Figs. 5 and 6. These figures are best understood as vertical and horizontal cuts through the phase diagram in Fig. 1, with the cuts chosen such that all features of the phase diagram can be understood and interpreted with the help of Figs. 5 and 6. We

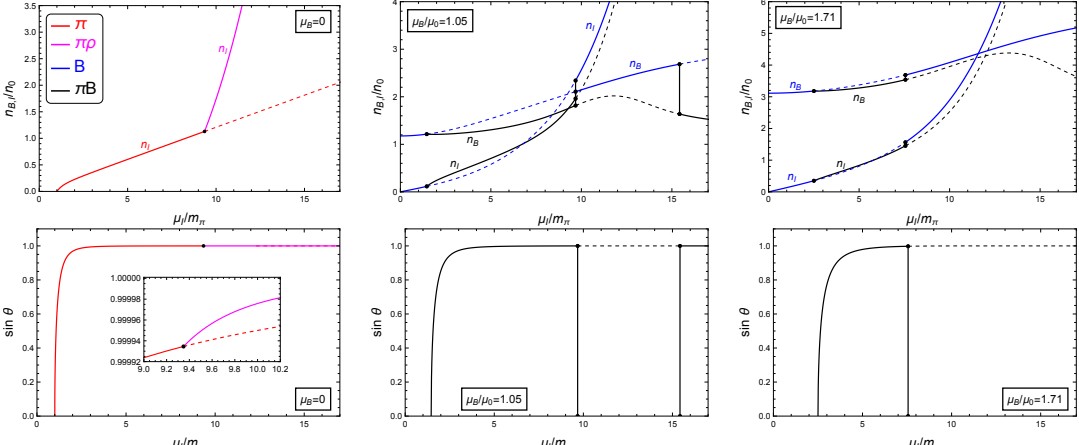

Figure 6: Same as Fig. 5, but as functions of the isospin chemical potential for three different values of the baryon chemical potential.

collect the main observations in the following list.

- *Thermodynamic consistency.* Following the stable phases for increasing $\mu_I$ ($\mu_B$), the corresponding density $n_I$ ($n_B$) is either constant or increases, as it should. This includes first-order phase transitions, where the densities increase discontinuously.

- *Onset of baryons.* At vanishing isospin chemical potential, the onset of baryons occurs at $\mu_0$, which, due to our use of the "mesonic fit" (121), assumes the unphysical value $\mu_0 \simeq 1430\,\text{MeV}$. All densities are given in units of the onset density at this point, which is $n_0 \simeq 0.43\,\text{fm}^{-3}$, also significantly different from the QCD value in this fit. The upper row in Fig. 5 shows that the onset chemical potential decreases with increasing $\mu_I$ and eventually the onset becomes second order. We also find that, for sufficiently small $\mu_B$ and any $\mu_I$, the baryonic phases are either metastable (B, blue curves) or do not even exist as a solution of the equations of motion ($\pi$B, black curves).

- *Pion condensate.* One of the novelties of our approach is the dynamical calculation of the pion condensate in the medium of holographic baryons. As discussed, in the $\pi$ phase our results merely reproduce chiral perturbation theory, but all other phases with pion condensate (including the transition from the $\pi$ phase to them) are a prediction of our holographic approach. The behavior is shown in the lower panels of Figs. 5 and 6. In Fig. 5 we observe that the pion condensate vanishes for sufficiently large baryon chemical potentials. In the lower left panel of Fig. 6 the inset shows the onset of rho meson condensation from the $\pi$ phase. Interestingly, the pion condensate is larger compared to its values in the (then metastable) pure pion-condensed phase. We also see in both Figs. 5 and 6 that the pion condensate is (either zero or) essentially maximal, $\sin\theta \simeq 1$, for $\mu_I \gtrsim 5\,m_\pi$. This confirms the observation of Fig. 3 that the massless limit is a good approximation in this regime.

- *Rho meson condensation.* We have interpreted a large anisotropy in our gauge fields as a sign of rho meson condensation in baryonic matter. We can confirm this interpretation now with the thermodynamic behavior. Let us first consider the baryon and isospin number densities in the B phase in the upper middle panel of Fig. 5 (blue curves). For small $\mu_B$, this phase is metastable, nevertheless the behavior of $n_B$ is instructive. We see that it is very small until it starts to increase rapidly at about $\mu_B \sim 0.3\mu_0$. This suggests that for small $\mu_B$ it is mainly rho mesons that account for isospin number and that there

is a rather sharp crossover to baryonic matter. For the $\pi$B phase, let us consider the densities in the upper middle panel of Fig. 6 (black curves). At large $\mu_I$ we find an interesting maximum of the baryon density. The system decides to remove baryons while the isospin density keeps increasing without qualitative change. Since the pion condensate is essentially maximal throughout this regime, the only possible interpretation is that a rho meson condensate now provides a large fraction of the isospin density. This corresponds to a large anisotropy of the gauge fields, cf. Fig. 4.

- *Disappearance and re-appearance of the pion condensate.* A surprising result in the phase diagram in Fig. 1 is the non-monotonic behavior of the first-order $\pi$B-B phase transition line. As a consequence, there is a region where, upon increasing $\mu_I$ at fixed $\mu_B$, the pion condensate disappears, before re-appearing at even larger $\mu_I$. This scenario is shown in the middle panels of Fig. 6 (the right panels would show the same behavior if the $\mu_I$ scale was extended to larger values). A possible interpretation is that the system switches off the pion condensate because rho mesons become the more efficient way to generate isospin number. And, as we know at least from the case without baryons, the rho meson mass is larger in the presence of a pion condensate. Hence the rho meson condensate is less costly without the pions. This interpretation is supported by the observation that once rho mesons contribute significantly to the isospin density in the $\pi$B phase (judging from the decrease in $n_B$), this phase is again favored, i.e., the pion condensate re-appears. While Fig. 3 suggests that the non-monotonicity is not an artifact of our diagonal ansatz, we should keep in mind that we always work in the probe brane approximation, which becomes questionable if the gauge fields on the flavor branes are large. Moreover, we have only considered the confined geometry and therefore our setup does not show a deconfinement or chiral phase transition, which is expected in QCD at very large chemical potentials. Also, we find that the non-monotonicity disappears if the coupling strength is increased, as we demonstrate in the next subsection.

## 5.5 Phase diagram at different coupling strengths

The purpose of this subsection is two-fold: firstly, we compare the phase diagrams of the "mesonic fit" (121) with the one from the "combined fit" (123); secondly, we study the phase diagram at large $\lambda$. Both aspects are covered by the four panels of Fig. 7.

The upper left panel shows the phase diagram of the combined fit. The most striking observation is the difference to the mesonic fit in the region of large isospin chemical potential. The $\pi\rho$ phase has become metastable and disappears from the phase diagram. Instead, there is a first-order transition from the $\pi$ phase to the pure baryonic phase. Since this transition occurs at relatively small baryon chemical potentials, the B phase contains a significant admixture of a rho meson condensate in this regime. At $\mu_B = 0$, this transition can thus be interpreted as a transition from the $\pi$ to the $\rho$ phase. In other words, at sufficiently small coupling $\lambda$, the rho condensate is not added on top of the pion condensate, but it is more favorable, beyond a critical $\mu_I$, to switch off the pion condensate and replace it by a pure rho condensate. Due to the unphysical $f_\pi$ in the combined fit we do not interpret this observation as a prediction for QCD. We expect the prediction of the mesonic fit to be more accurate in this regime because at $\mu_B = 0$ there are no baryons and the shortcomings of this fit are not relevant. At small $\mu_I$, on the other hand, we observe that the phase structure is identical for the two fits. In particular, in both cases neutron stars exist only in the B phase, i.e., the conclusion that there is no pion condensation under neutron star conditions does not depend on the fit. Therefore, the mesonic fit captures all our (qualitative) QCD predictions, and thus it was used for the main results in Sec. 1.2.

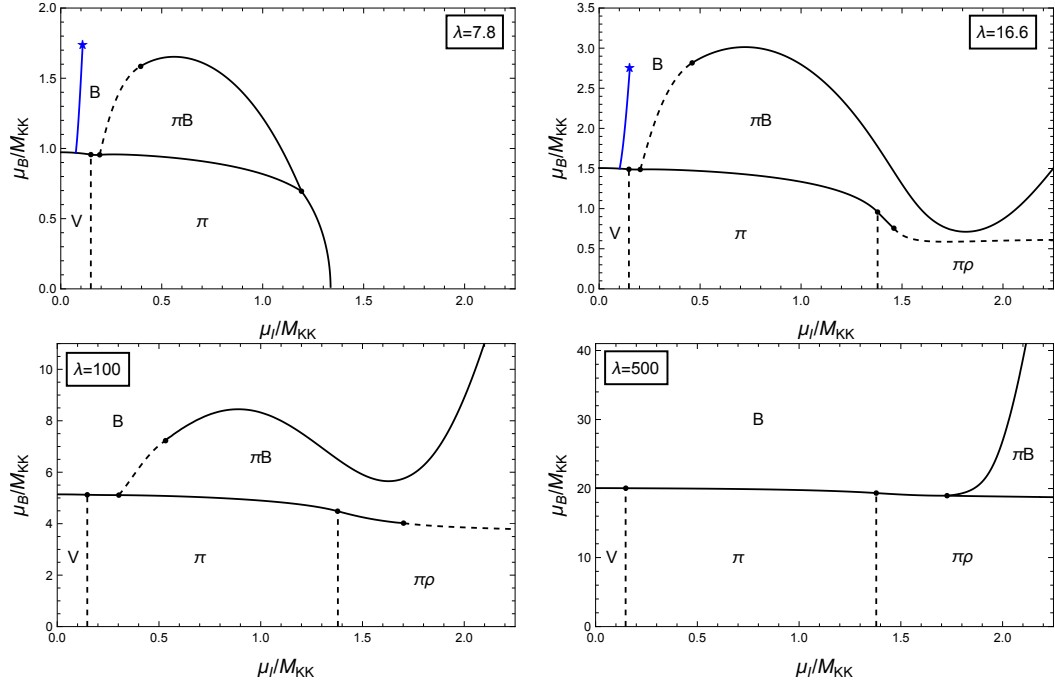

Figure 7: Phase diagram for four different values of the 't Hooft coupling $\lambda$. The upper panels correspond to the "combined fit" [left, see Eq. (123)] and the "mesonic fit" [right, see Eq. (121), used for Fig. 1] and include neutron star matter (blue curves, endpoints corresponding to the center of the most massive star). In each case, $\bar{m}_\pi$ is chosen such that the ratio $m_\pi/m_\rho$ is equal to its physical value.

It is also instructive to study the structure of the phase diagram for very large values of $\lambda$. Although irrelevant for fitting basic properties of real-world baryons or mesons, this is the limit in which our classical gravity approximation is valid, while small values of $\lambda$ (and $N_c$) rely, strictly speaking, on uncontrolled extrapolations. We have not found a direct way to compute the $\lambda \to \infty$ version of our phase diagram, but we have re-calculated the phase transition lines for the two additional values $\lambda = 100$ and $\lambda = 500$. By showing the results in units of $M_{\text{KK}}$, this requires no specific choice of the Kaluza-Klein scale. However, the dimensionless pion mass parameter $\bar{m}_\pi$ cannot be scaled out of the equations and we need to decide how to readjust it upon variation of $\lambda$. From Eq. (99) we know that the rho meson mass $m_\rho$ scales with $\lambda^0$, while the pion mass $m_\pi$ scales with $\lambda$ for fixed $\bar{m}_\pi$, see Eq. (37). Hence, if we kept $\bar{m}_\pi$ fixed, the rho meson would become lighter relative to the pion as we increase $\lambda$ (and actually lighter than the pion for sufficiently large $\lambda$). For a sensible comparison, we therefore adjust $\bar{m}_\pi$ such that the ratio $m_\pi/m_\rho$ keeps its physical value. This results in $\bar{m}_\pi \simeq 0.019, 0.0037$ for $\lambda = 100, 500$. The phase diagrams thus computed are shown in the lower panels of Fig. 7. As a consequence of keeping the meson mass ratio fixed and choosing the isospin chemical potential in units of the Kaluza-Klein scale as the horizontal axis, the $V \to \pi$ and $\pi \to \pi\rho$ transitions (if they exist) occur at the same critical points for all values of $\lambda$. The baryon onset at vanishing isospin chemical potential scales with the baryon mass (minus the binding energy of symmetric nuclear matter at saturation, which is different for each $\lambda$). Since the baryon mass scales with $\lambda$ [74], the critical baryon chemical potential for the $V \to B$ transition increases from the upper left to the lower right panel.

The first main observation is that the phase transition lines that separate the purely mesonic phases from the phases containing baryons becomes more and more horizontal as the coupling strength is increased. In other words, the properties of the baryons seem to become essentially

independent of the isospin chemical potential. This observation is in accordance with the properties of a single baryon at large $\lambda$. In this case, the BPST solution is, to leading order, unaffected by isospin (and baryon) chemical potentials, which simply introduce non-trivial – sub-leading – temporal components of the gauge fields, see appendix A of Ref. [39]. We also see that the curious non-monotonicity of the B-$\pi$B transition disappears gradually for large $\lambda$. As a consequence, in the strongly coupled limit pion condensation in baryonic matter is restricted to a region of extremely large isospin chemical potentials. This is an interesting observation also for neutron star applications: if this tendency is of general value, the absence of a pion condensate in dense nuclear matter – which we observe even for the physical, less strongly coupled parameter set – may be interpreted as a strong-coupling effect.

# Acknowledgments

We thank Mark Alford, Lorenzo Bartolini, Christian Ecker, Nick Evans, Sven Bjarke Gudnason, Matti Järvinen, and Jack Mitchell for valuable discussions.

**Funding information** The work of N.K. is supported by the ERC Consolidator Grant 772408-Stringlandscape. A.P. is supported by the National Research Foundation of Korea under the grants, NRF-2022R1A2B5B02002247, NRF-2020R1A2C1008497. N.K. and A.P. acknowledge the hospitality of the APCTP in Pohang, South Korea, where part of this work was developed.

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
