# Peer review of "Phases of cold holographic QCD: baryons, pions and rho mesons"

_SciPost Physics, doi:SciPost Phys. 15, 162 (2023)_

## Round 1 · Referee Report · Anonymous (Referee 2) · 2023-5-17

Report

The paper analyzes in details the phase diagram in the baryon and isospin chemical potential plane of the holographic QCD model of Sakai and Sugimoto in the confined phase. The paper is very well written, the computations and their implications are clearly stated and commented. A number of results are new and interesting. I believe they are worth to be published in SciPost. Nevertheless, I believe the presentation could be more reader-friendly if the following points are considered. 1. The authors discuss in details some limitations of their approach, for example the use of a diagonal ansatz for the gauge fields: they find out that it is reasonable to assume that this crude approximation gives a qualitatively correct phase diagram - I tend to agree with this statement. Nevertheless, as far as I can see, this issue does not appear anywhere in the "Introduction and conclusions" section. Being the latter quite long, and being absent a concluding section, mentioning in the Introduction the used approximation would be fair to the reader. 2. I would also appreciate a line or two in the Introduction section on the dependence of the phase diagram on the parameter \lambda, discussed in section 5.5. In fact, the phase diagram of figure 1 is calculated for a relatively small value of \lambda (smaller than 8). For larger values, as can be seen in figure 7, the phase diagram presents some qualitative difference. Since the gravity computations are reliable in the large \lambda limit, this could point towards the fact that some features of the phase diagram in figure 1 could change by including 1/\lambda corrections.

---

## Round 1 · Referee Report · Anonymous (Referee 1) · 2023-5-17

Report

This manuscript analyzes in detail the phase structure of QCD at finite baryon and isospin chemical potentials by employing the holographic Witten-Sakai-Sugimoto model. This study is continuation of earlier work in this topic. The authors extend earlier studies by considering the effects of finite pion mass and the effect of rho meson condensation, which leads to a complex phase diagram containing several interesting new results. The study is probably the most extensive exploration of the phases of holographic QCD at finite isospin chemical potential. It is also well motivated, as not so much is known about the phase structure of QCD at high densities from first principle QCD analysis, and some of the phases are relevant for neutron star physics. Due to the complexity of the phase structure the article is quite technical, but the details are explained very well. Therefore I think that the manuscript clearly passes the acceptance criteria for SciPost Physics. I have only minor comments and questions that the authors should check.

On page 10 below Eq. (25), it is written "...the non-zero field strengths are...", but the expressions which are referred to here are not field strengths but gauge field components.

On page 11 below Eq. (30), "Subtracting this vacuum contribution, the factor from the Nambu-Goto action is 1." I did not understand this comment, do the authors mean that $e^{-S_{NG}}$ evaluates to one?

On page 17, the authors discuss the equations of motion. They refer to the "full" equations of motion in Eqs. (29). However in
Eq. (50) an approximate Lagrangian density was introduced. Do I understand correctly that the equations of motion on page 17 match with the "diagonal" components of Eqs. (29), and are also consistent with the approximate action of Eq. (50)?

On page 18 below Eq. (67) the authors comment "This result ... does assume its derivative to be discontinuous." But if $h$ is odd, as far as I can see the derivative is, in fact, continuous (while the function itself is discontinuous).

In section 4.5 the authors discuss the $\beta$-equilibrium and charge neutrality conditions for neutron star matter. Maybe it would be a good idea to remind at the end of this section what they are used for, as at the moment the text ends rather abruptly. Do I understand correctly that the neutron star matter conditions are only used to draw the blue line in Fig. 1, but are otherwise irrelevant for the phase diagram?

---

## Round 2 · Referee Report · Anonymous (Referee 2) · 2023-7-6

Report

I recommend the paper for publication

---

## Round 2 · Referee Report · Anonymous (Referee 1) · 2023-7-31

Report

I thank the authors for the detailed response. The main change with respect to v1 is the inclusion of the spatial Abelian components of the gauge fields, and as far as I can see this is done consistently. Therefore I recommend the article for publication.

I spotted one typo: in Eq. (55) $e_i$ has been eliminated, but it is still mentioned in the text after the equation.

---

## Round 2 · Author Response

We thank both referees for their positive reports and their helpful questions and suggestions. While preparing our modified manuscript we realized that in our ansatz for the gauge fields we made an unjustified assumption by omitting the spatial abelian components. We have therefore included these components now, resulting in a significantly revised manuscript but with main conclusions unchanged, such that we think the new manuscript can be considered as a resubmission rather than a completely new submission. Let us explain our improvements:

(i) The structure of the paper has remained intact: the titles and order of all sections and subsections is the same as before.

(ii) All main results are the same as before for a somewhat nontrivial reason: Including the spatial abelian components does change the numerical results but for reasons now explained in Sec 1.2 we had to adjust the model parameters and due to this adjustment the main conclusions are unchanged. As a consequence, all plots that appeared before also appear now, with all curves adjusted but all main observations unchanged. The only difference in presentation of the results is Fig 7, which now contains 4 instead of 3 panels.

(iii) It would have been too tedious to present a list of all changes in the text or to highlight them since small but numerous adjustments had to be made in several sections. Let us instead sketch the main changes here: The list in Sec 1.2 contains a new item "Parameter dependence" to explain the point already mentioned in (ii). This item also contains a new footnote mentioning the difference to the previous version. Many equations in Secs 2.2.1, 3.1, and 3.2 had to be adjusted due to the new nonzero gauge field components. These changes are largely straightforward, starting from the extended Lagrangian in Eq. (28). In particular, the equations of motion of the new components can be solved analytically and thus simply give rise to extra terms in the other, already existing equations, rather than inducing additional nontrivial equations. Sec 5.5 has changed significantly, now containing a first part discussing two different physical parameter fits, which was absent before. All other sections are largely the same as before, except for various necessary but minor adjustments in the text (for instance the discussion of the different phases in Sec 4 is basically unchanged except for additional terms in the equations).

Let us next give our replies to the referee's comments and the resulting additional changes to the manuscript:

Referee 1

1) We have included an additional paragraph in the introduction (towards the middle of page 3), pointing out our approximation of the diagonal ansatz, which reads

"Nevertheless, even in our generalized ansatz we employ a simplification by keeping the non-abelian gauge fields diagonal – locking spatial indices with those indicating the orientation in isospin space. We will argue that this is a good approximation for our purposes since our main results are tightly constrained by the regimes where the diagonal ansatz does provide an exact solution to our equations of motion."

2) We have added a remark regarding the value of the 't Hooft coupling in the introduction/conclusion section above the itemized list on page 4:

"As in most previous applications of the model, this implies an extrapolation from the regime of very large \lambda, where the classical gravity approximation in the bulk is valid, down to smaller, but not perturbatively small, coupling strengths."

Moreover, as already mentioned above, we have added a new item "Parameter dependence" on pages 5/6. We have also slightly extended our already existing comment on the dependence of the phase diagram on lambda in the item before that on page 5, which now reads:

"We will discuss the dependence of the phase structure on the ’t Hooft coupling \lambda and see that the non-monotonicity disappears for large values of \lambda, where our classical gravity approximation is more reliable."

Referee 2

1) We thank the referee for pointing out this inaccuracy. Since with our extended ansatz the sentence below eq (25) is no longer necessary in its previous form this inaccuracy is gone.

2) Yes, the referee understood this correctly. We have made this statement more precise by now saying "...the exponential containing the Nambu-Goto action is 1."

3) Yes, if the equations of motion are derived from eq (51) (former eq (50)) one would in principle obtain the equations of motion on p18 (p17 in the previous version), provided one ignores the off-diagonal components and their own equation of motion. However, the equation of motion for h on p18 contains an additional constraint, which we hadn't explained clearly in the previous version. Namely, to respect the symmetry of the system in the presence of the chiral transformation in the boundary conditions (45), we need to work with the constraint h_1=h_2, which is now explained around eq (60).

4) We thank the referee for pointing out this inaccuracy. Eq (71) (former eq (67)) is correct also for the case of a continuous derivative, in contrast to what our statement suggested. We have deleted this misleading statement.

5) We have followed the suggestion of the referee and clarified at the end of sec. 4.5 that the conditions of beta-equilibrium and charge neutrality are only used to calculate the blue line in Fig 1. We added the sentence

"We perform this calculation to indicate the location of beta-equilibrated, charge neutral matter in the phase diagram (blue lines in Figs. 1 and 7); for all other results \mu_B and \mu_I are independent and the results of this subsection play no role."

---

## Round 3 · Author Response

Small typo fixed.

---

## Editorial Decision

published